# Fibrillin microfibril structure identifies long-range effects of inherited pathogenic mutations affecting a key regulatory latent TGFβ-binding site

Alan R. F. Godwin[1], Rana Dajani[1], Xinyang Zhang[1], Jennifer Thomson[1], David F. Holmes[1], Christin S. Adamo [2,3], Gerhard Sengle[2,3,4,5], Michael J. Sherratt[6], Alan M. Roseman [7] & Clair Baldock [1] ✉

Genetic mutations in fibrillin microfibrils cause serious inherited diseases, such as Marfan syndrome and Weill–Marchesani syndrome (WMS). These diseases typically show major dysregulation of tissue development and growth, particularly in skeletal long bones, but links between the mutations and the diseases are unknown. Here we describe a detailed structural analysis of native fibrillin microfibrils from mammalian tissue by cryogenic electron microscopy. The major bead region showed pseudo eightfold symmetry where the amino and carboxy termini reside. On the basis of this structure, we show that a WMS deletion mutation leads to the induction of a structural rearrangement that blocks interaction with latent TGFβ-binding protein-1 at a remote site. Separate deletion of this binding site resulted in the assembly of shorter fibrillin microfibrils with structural alterations. The integrin $\alpha_v\beta_3$-binding site was also mapped onto the microfibril structure. These results establish that in complex extracellular assemblies, such as fibrillin microfibrils, mutations may have long-range structural consequences leading to the disruption of growth factor signaling and the development of disease.

Elastic fibers are essential components of all mammalian elastic tissues, such as blood vessels, lung and skin[1]. Elastic fiber formation requires fibrillin microfibrils to act as a scaffold for the deposition of tropoelastin. These microfibrils also provide limited elasticity to elastin-free tissues, such as the ciliary zonule, a ligament essential for lens attachment in the eye. In many tissues, these assemblies provide a multifunctional platform for the interaction of matrix molecules required for elastic fiber assembly and function, and mediate cell attachment[2]. Fibrillin is needed for the correct assembly of many microfibril-associated proteins, including members of the latent transforming growth factor β (TGFβ)-binding protein (LTBP) family[2].

The fibrillin superfamily is composed of fibrillins (isoforms 1–3), with fibrillin-1 being predominant in adults[3–6], and the four structurally related LTBPs (isoforms 1–4)[7–10]. Fibrillin superfamily proteins contain

[1]Wellcome Trust Centre for Cell-Matrix Research, Division of Cell Matrix Biology and Regenerative Medicine, School of Biological Sciences, Faculty of Biology, Medicine and Health, University of Manchester, Manchester Academic Health Science Centre, Manchester, UK. [2]Center for Biochemistry, Faculty of Medicine and University Hospital Cologne, University of Cologne, Cologne, Germany. [3]Department of Pediatrics and Adolescent Medicine, Faculty of Medicine and University Hospital Cologne, University of Cologne, Cologne, Germany. [4]Center for Molecular Medicine Cologne, University of Cologne, Cologne, Germany. [5]Cologne Center for Musculoskeletal Biomechanics, Cologne, Germany. [6]Division of Cell Matrix Biology and Regenerative Medicine, School of Biological Sciences, Faculty of Biology, Medicine and Health, University of Manchester, Manchester Academic Health Science Centre, Manchester, UK. [7]Division of Molecular and Cellular Function, School of Biological Sciences, Faculty of Biology, Medicine and Health, University of Manchester, Manchester Academic Health Science Centre, Manchester, UK. ✉e-mail: clair.baldock@manchester.ac.uk

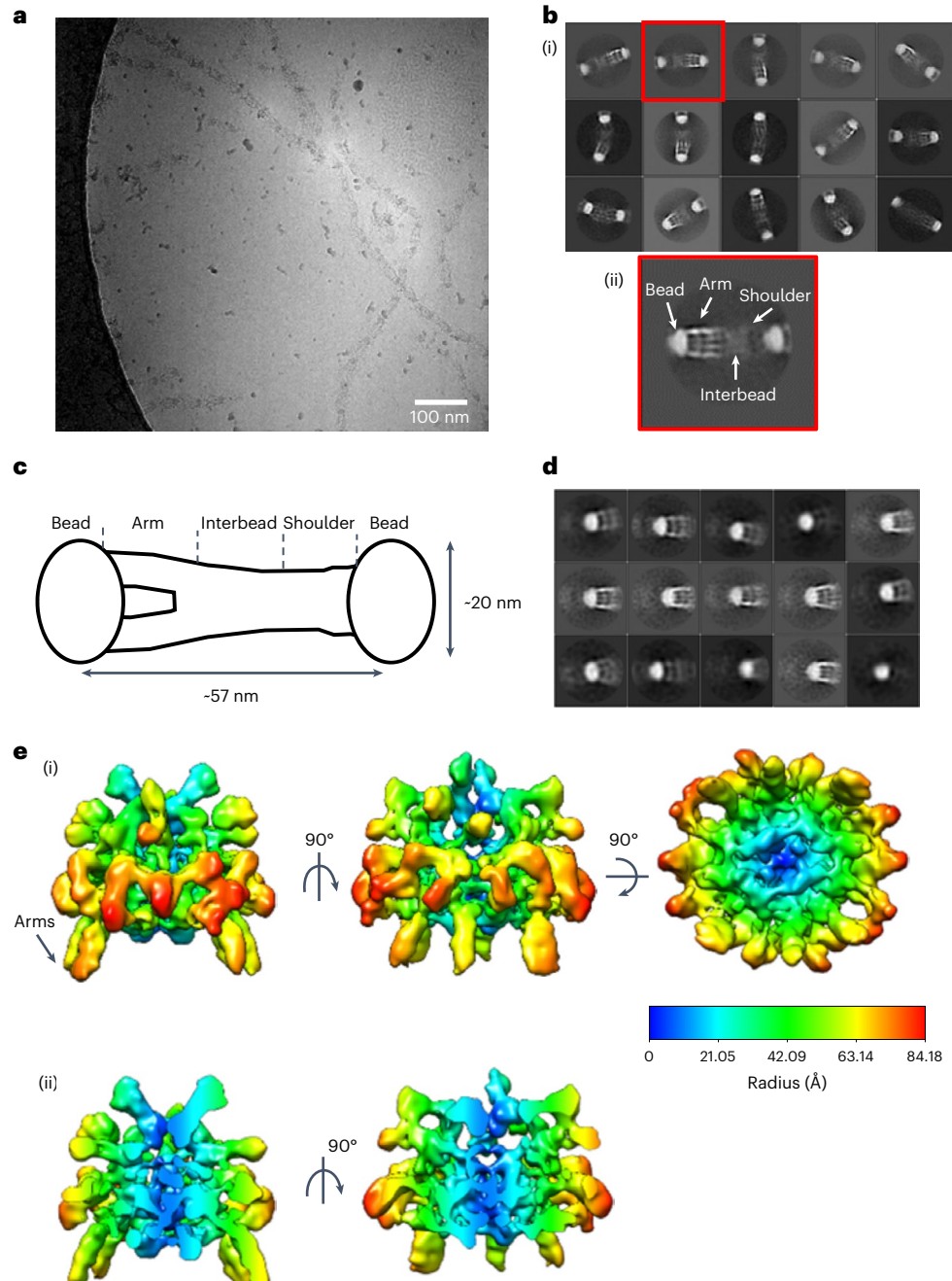

**Fig. 1 | Cryo-EM structure of the fibrillin microfibril bead region.** Fibrillin microfibrils were extracted from bovine ciliary zonules under nondenaturing conditions and imaged using cryo-EM. **a**, Representative cryo-EM image of purified bovine ciliary zonule microfibrils, from a dataset of 1,310 images. Scale bar, 100 nm. **b**, (i) Reference-free classification of the full fibrillin microfibril repeat. Box size, 100 nm. (ii) A class highlighted in panel (i) by a red box was rotated 180° and enlarged to highlight the different microfibril regions.

**c**, Schematic of the fibrillin microfibril, highlighting the bead, arm, interbead and shoulder regions of the microfibril. **d**, Classification of particles aligned to the fibrillin bead region. Box size, 57 nm. **e**, (i) Cryo-EM single-particle reconstruction of the fibrillin bead region shown in three orthogonal orientations. The bead is rainbow colored by cylindrical radius from blue (at the center of the bead) to red (on the outside of the bead). (ii) Two orthogonal views of the bead reconstruction have been sliced to show a cross-section through the center of the bead.

arrays of epidermal growth factor-like (EGF) domains interspersed with TGFβ-binding like (TB) domains and hybrid domains[3]. Of the 47 EGF domains in fibrillin, 43 are calcium binding (cb)[3]. There are seven TB domains (also referred to as 8-cysteine domains) that are unique to the fibrillin superfamily.

Fibrillin assembles to form beaded microfibrils with ~57 nm periodicity[11] and a mass of ~2.55 MDa per repeat[12]. These microfibrils are polar polymers that are formed by linear assembly of fibrillin molecules via direct interactions between the amino (N) and carboxy (C) termini[13].

Lateral association of fibrillin molecules also occurs and is driven by a homotypic interaction between these termini to form mature microfibrils[14–16]. Microfibril structure can be described by four distinct regions based on banding pattern: the bead, arms, interbead and shoulder regions (Fig. 1c)[17,18]. However, the organization of fibrillin molecules into mature microfibrils and their three-dimensional (3D) structure is still unclear, as their complexity and variability prevent analysis by most structural biology and biochemical techniques. Therefore, a number of packing models have been proposed, including an intramolecular

pleating model (where each fibrillin molecule spans one 57-nm period[19]) and a one-half staggered model (where each fibrillin molecule spans two 57-nm periods[20]). Both models suggest that the N and C termini are located near the bead region.

In addition to their role in elastic fiber assembly, fibrillin microfibrils mediate tissue homeostasis, which when perturbed by fibrillin mutations causes a number of heritable connective tissue disorders, such as Marfan syndrome and Weill–Marchesani syndrome (WMS)[21]. This tissue homeostasis is mediated via fibrillin interactions with cell-surface receptors (such as integrins[22–25] and heparan sulfate proteoglycans[26]) and with growth factors (including TGFβ and bone morphogenetic proteins (BMPs)[27–29]). Of the four LTBP isoforms, LTBP-1, -3 and -4 play important roles in the processing and secretion of TGFβ[30], through covalent binding to the latency-associated peptide of TGFβ, producing the large latent complex (LLC) of TGFβ[31,32]. The LLC becomes sequestered within the matrix and can regulate TGFβ bioavailability[31–33]. Fibrillin is thought to have a role in the regulation of latent TGFβ, as the first hybrid domain in fibrillin binds to the C-terminal region of LTBP-1 (ref. 27) and dysregulated TGFβ signaling is a characteristic of Marfan syndrome[34]. However, mice homozygous for deletion of the fibrillin-1 hybrid1 domain are able to form normal microfibrils[35]. A common WMS alteration is a three-domain fibrillin-1 deletion in an adjacent region (domains TB1–proline-rich region (PRR)–EGF4). When replicated in mice, this deletion resulted in a WMS-like phenotype, with thick skin, short stature and brachydactyly[36].

Although growth factor binding and disease-linked sites have been mapped in vitro using recombinant fibrillin-1 fragments, their exact sites of interaction on the microfibril are unknown. Most microfibril-binding proteins, including proBMP complexes, LTBPs (and the LLC), fibulins, microfibril-associated glycoproteins (MAGPs) and a disintegrin and metalloproteinase with thrombospondin motifs (ADAMTS) and ADAMTS-like proteins bind near to the N-terminal region of the fibrillin monomer (for a review, see ref. 2). However, it is unknown how multiple binding partners can interact with microfibrils and whether this binding is competitive. Previous negative stain electron microscopy studies have not achieved sufficient resolution (~43 Å) to determine the arrangement of fibrillin molecules in tissue microfibrils[37]. To address this issue, we have taken advantage of the recent advances in cryo-EM to determine the structure of fibrillin microfibrils extracted from mammalian tissue to subnanometer resolution. This structure allows the tracking of individual fibrillin molecules through the bead and arm regions. To confirm the location of important functional regions within the microfibril, microfibrils from mouse models containing deletions of the hybrid1 domain that binds latent TGFβ, as well as the WMS-causing deletion, were analyzed. The resultant structural perturbations, together with binding analyses and scanning transmission electron microscopy (STEM) mass mapping further defined the fibrillin molecular organization and the locations of the LTBP-1- (and LLC-) and integrin-binding sites.

## Results

### 3D reconstruction of the fibrillin microfibril bead region

To determine the 3D structure of the fibrillin microfibril repeat, microfibrils were purified from bovine ciliary zonules—a pure source of fibrillin microfibrils—using sonication and size exclusion chromatography and imaged using cryo-EM (Fig. 1a). When imaged, the microfibrils had the characteristic beads-on-a-string appearance[37,38]. A total of 27,737 microfibril repeating units were digitally extracted from the images and classified into 2D classes (Fig. 1b). As previously described, some heterogeneity was apparent in the class averages due to flexibility along the microfibril axis[37]. To overcome this, and with the objective of increasing the resolution of the microfibril reconstruction, separate submodels of specific microfibril regions (that is, the bead and arm regions) were created, by centering the reconstruction locally on each of these regions. Particles that had been aligned to the full fibrillin

### Table 1 | Cryo-EM data collection statistics

| | Bead (EMDB-13984) | Arm (EMDB-13986) |
|---|---|---|
| Magnification | 64,000× | 64,000× |
| Voltage (kV) | 300 | 300 |
| Electron exposure (e⁻ Å⁻²) | 66 | 66 |
| Defocus range (µm) | 2–5 | 2–5 |
| Pixel size (Å) | 2.22 | 2.22 |
| Symmetry imposed | C2 | C2 |
| Initial particle images | 27,737 | 27,737 |
| Final particle images | 7,139 | 4,957 |
| Map resolution (Å) | 9.7 | 18.3 |
| FSC threshold | 0.143 | 0.143 |
| Map resolution range (Å) | 4.6.0–36.7 | 13.6–24.2 |

The EMDB accession codes for the fibrillin bead and arm regions are EMD-13984 and EMD-13986, respectively. FSC, Fourier shell correlation.

repeat were recentered to locate the bead coordinates at the particle center. Classification centered on the bead region yielded classes with the characteristic bead, with arms emerging from one side of the bead (Fig. 1b,d). The twofold symmetry previously observed in the fibrillin structure[37] was confirmed by examination of an initial unsymmetrized reconstruction (Extended Data Fig. 1a). Twofold symmetry was applied along the microfibril axis and a 3D reconstruction of the bead region was generated with 7,139 particles used in the final refinement. The resolution of the microfibril bead region (9.7 Å) was estimated using Fourier shell correlation of two independently refined half-maps and applying the 0.143 criterion (Table 1 and Extended Data Fig. 1b). The bead shape is approximately spherical; it can be enclosed by an ellipsoid with dimensions of 16.5 nm × 15 nm × 12 nm and has a complex interwoven arrangement (Fig. 1e). A mask around the bead was used in the processing, but despite not being under the mask, the arm region could still be seen protruding from the bead. The resolution allows the individual fibrillin molecules to be visualized interwoven through the bead. We believe that this structure represents a core of fibrillin molecules. Any microfibril-associated proteins would be present at lower stoichiometry and would not be detected in the overall reconstruction as it represents an average of many thousands of fibrillin repeats. MAGP1, which is typically present, is probably not detected due to its small size and unstructured nature[39]. The tight packing within this region supports data that show resistance of the microfibril bead region to proteolysis[40].

### Identifying protease-resistant domains in the bead region

The binding sites of two monoclonal antibodies (mAbs) raised against the N-terminal half of fibrillin have previously been mapped on the microfibril structure[13,38]. mAb1919 binds within the arm region, whereas mAb2502 binds on the side of the bead and specifically recognizes human microfibrils. Using short recombinant human fibrillin constructs[19], we narrowed down the antibody-binding sites by western blotting (Extended Data Fig. 2a,b). The epitope for mAb2502 was refined considerably as it detected two overlapping fibrillin constructs that restricted the epitope to within domains TB1 and the PRR. mAb1919 detected a construct containing domains cbEGF7–hybrid2 immediately downstream of TB2 (Extended Data Fig. 2b). Taken together, these data suggest that TB1 and/or the PRR are located on the peripheral edge of the bead, whereas domains downstream of TB2 are found in the arm/interbead region. Our recent mass spectrometry data identified a protease-resistant fibrillin region comprizing domains EGF4–TB2 (ref. 41). Peptides from this region were never detected by liquid chromatography with tandem mass spectrometry after elastase digestion

of microfibrils from skin, fibroblast cultures or ciliary body (Extended Data Fig. 2c). These findings are consistent with other proteomics studies on microfibrils from ciliary zonules digested with trypsin, where few peptides from this region were identified[42,43]. Taking this into account, together with the antibody epitope mapping data, we suggest that the EGF4–TB2 domains could be buried in the bead core and thus protected from proteolytic digestion (Extended Data Fig. 2d).

### Modeling the terminal regions into the fibrillin bead

The cryo-EM reconstruction was analyzed for regions that were buried in the core of the interwoven bead structure. A feature was observed in the center of the bead map that remained connected at high threshold levels (Fig. 2a). This feature could be traced through the center of the bead and was found to have a persistence length of 17 nm (measured with UCSF Chimera using volume tracer), where a diameter of ~2 nm was consistent with the width of the EGF and TB domains and the length supported a chain of around seven domains. Within the bead, there are four nonsymmetric copies of this chain (Fig. 2b), which appear twice due to the twofold symmetry, confirming the previously seen pseudo eightfold symmetric arrangement of eight fibrillin monomers per repeat within the fibrillin microfibril[37,44]. Three of these features are of similar dimensions but the fourth is shorter. This may be due to loss of density in the averaging caused by the flexibility of the arms. Moreover, the arm region was not under the 3D mask during refinement of the bead. The interface between the bead and arm could be seen more clearly in a lower-resolution reconstruction in which a larger mask was used for refinement (Extended Data Fig. 3). Guided by the antibody mapping data and starting at the edge of the bead, domain TB1, followed by the consecutive domains from PRR to cbEGF6, is docked into the density for one of these regions. The density where domains cbEGF5 and cbEGF6 are docked is contiguous with the density for the arm regions, giving confidence to these domain assignments. A small-angle X-ray scattering (SAXS)-derived model of this region was used to aid this docking (Fig. 2c (i)). The arrangement of these domains in the cryo-EM structure is very similar to the model based on experimental SAXS data[45].

Due to the dense molecular packing in the bead region and lower local resolution in the bead core region (Extended Data Fig. 1d), it is possible that the molecules in the core of the bead take a different path through the center of the bead. Antibody binding data have previously shown that the C-terminal region of fibrillin is also located near to the bead[13,38] and the C-terminal half of fibrillin-1 has been shown to self-assemble into bead-like multimers that have been suggested to initiate microfibril assembly[16]. Furthermore, there are differences in the predicted location of the N-terminal region in microfibrils purified from adult human ciliary zonule and human neonate foreskin, suggesting that the mab2502 (clone 26) epitope may be conformationally variable[13,38]. Therefore, we modeled an alternative path through the bead core, where the central core may be composed of N- and C-terminal interacting regions (Fig. 2c (ii)), consistent with antibody mapping data. Importantly, domains cbEGF5 and cbEGF6 would remain in the same location in both scenarios, as these domains connect and are contiguous with the density that continues into the arm region.

After accounting for the eight copies of this central region within the core of the microfibril bead (Fig. 2d), the remaining bead structure forms a ring around the outside surface and interwoven on the underside of the bead, as shown in gray in Fig. 2e. The approximate mass of the bead region estimated from the cryo-EM electron density map is ~1.1 MDa, which is consistent with STEM mass mapping. After docking eight symmetric copies of these seven fibrillin domains into the bead core, there is still sufficient mass and density to accommodate a further 12 or so fibrillin domains for each of the eight symmetric copies (supported by the eightfold pseudosymmetry). This suggests that there is quite an extensive overlap of N- and C-terminal regions within the bead.

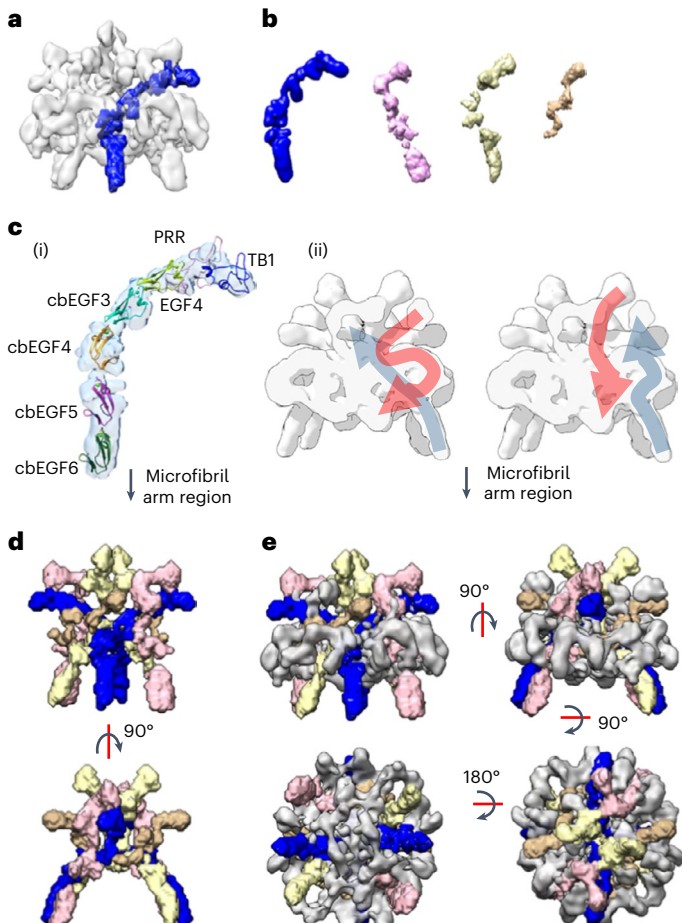

**Fig. 2 | Modeling of the fibrillin N-terminal region into the fibrillin bead region. a**, Cryo-EM structure of the fibrillin bead region with a feature (in the centre of the bead map) that remained connected at high threshold levels segmented and highlighted in blue. **b**, The four unique (not symmetry-related) copies of this region are segmented from the bead map. The feature shown in **a** is colored blue. As twofold symmetry has been applied to the reconstruction, there are also symmetric pairs of each of these features to give eight in total in the bead core. **c**, (i) A SAXS-derived model of the region encompassing TB1, the PRR, EGF4 and cbEGF3–6 (ref. 45) is docked into the region from the core of the bead segmented in **a** (shown as pale blue density). (ii) A cross-section through the bead density is shown with schematic representation of the N-terminal region (blue) running through the core of the bead (as modeled in **c** (i)) and C-terminal region (red) in the outer density. On the right, an alternative representation is presented where the C-terminal region could be in the inner core of the bead with the N-terminal region around the outside. In both, the cbEGF5 and cbEGF6 domains would be in the same location as their connection to the density in the arm region is contiguous. **d**, The four central densities shown in **b** and their symmetric pairs are segmented from the bead structure and colored as in **b** and are shown without the remaining bead structure. **e**, As in **d**, but shown with the remaining bead density colored gray.

### Mutated microfibrils reveal a disrupted shoulder region

After locating the domains downstream of TB1 in the bead region, we predicted that the upstream N-terminal domains would be in the shoulder region. To confirm this, microfibrils from two fibrillin mouse models with either a domain deletion upstream, or downstream including the TB1 domain, were analyzed. Imaging these microfibrils allowed us to confirm the positions of these domains in the microfibril reconstruction and to analyze the structural consequences of deletion of fibrillin domains on microfibril structure. Microfibrils were purified from skin from 6-week-old mice homozygous for either a deletion of the first hybrid domain (ΔH1) (ref. 35); a domain that contributes

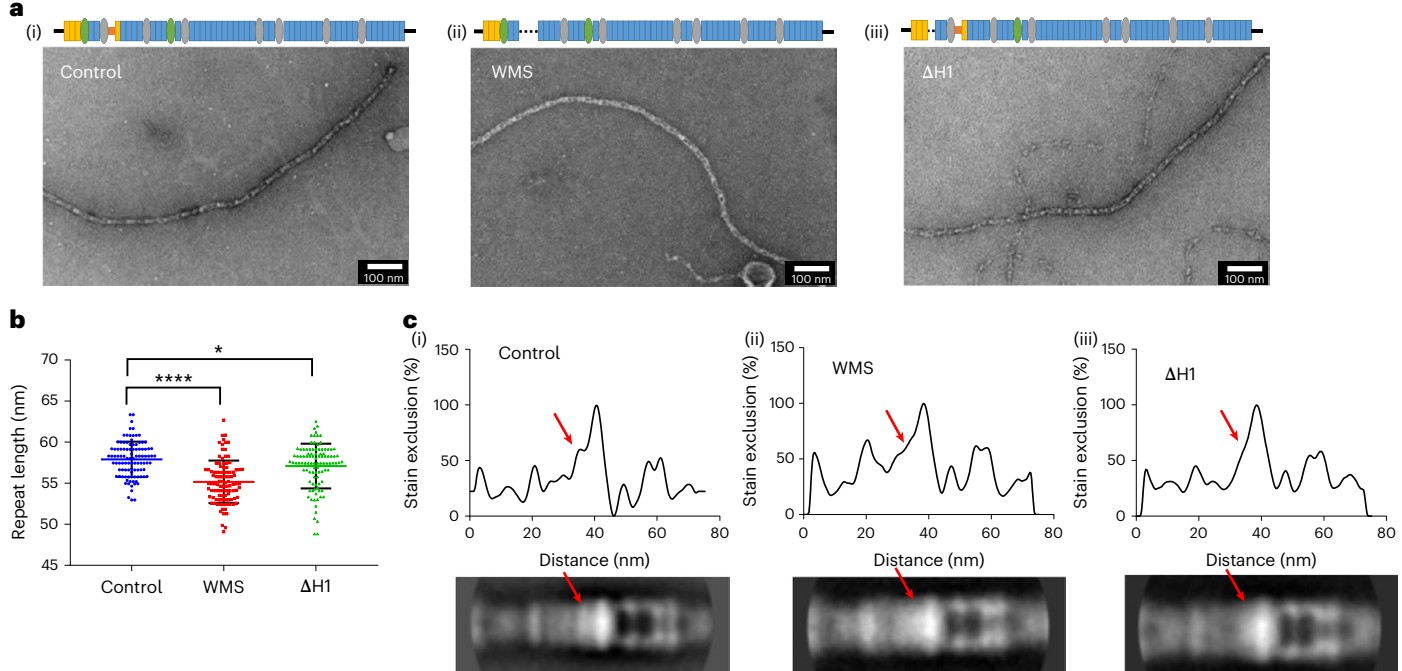

**Fig. 3 | Mutant fibrillin microfibrils reveal a disrupted shoulder region.**
Fibrillin microfibrils extracted from control and homozygous WMS and ΔH1 mouse skin and imaged using negative stain TEM. **a**, Representative TEM images of (i) control, (ii) WMS and (iii) ΔH1 extracted fibrillin microfibrils. Scale bars, 100 nm. Schematic of the fibrillin domain structure is shown above each image, with the WMS and ΔH1 domain deletions represented as dotted lines. **b**, Scatter plot showing the periodicities of the control, WMS and ΔH1 extracted fibrillin microfibrils. The data represent means ± s.d. For each dataset, 100 microfibril periods were measured and statistical analysis shows

that the mutated microfibrils were significantly different from the control (*$P$ = 0.042; ****$P$ < 0.0001), as determined by one-way analysis of variance with Dunnett's multiple comparisons test. **c**, Averaged images were generated from 458 wild-type periods, 472 ΔH1 periods and 476 WMS periods. The top panels show 2D plots of the stain exclusion/intensity profile across the averaged fibrillin microfibril period (bottom panels) for (i) control, (ii) WMS and (iii) ΔH1 microfibrils. The red arrows highlight a region at one side of the bead of the microfibril that is disrupted in the mutated microfibrils.

to the fibrillin-binding site for latent TGFβ via LTBP-1 (ref. 27) or a deletion of three domains (TB1–PRR–EGF4) that causes WMS[36]. As microfibrils are more difficult to extract from skin, negative staining electron microscopy was used to image microfibrils, as it has a much higher signal-to-noise ratio than cryo-EM and can gain more information from fewer microfibrils. A total of 100 microfibril periods were measured for each dataset and compared with wild-type microfibrils (Fig. 3a,b). The control microfibrils had a periodicity of 57.95 ± 0.22 nm. The ΔH1 microfibrils had a periodicity of 57.14 ± 0.27 nm, which was 0.81 nm shorter than that of the control. The WMS microfibrils had a periodicity of 55.21 ± 0.26 nm, which was 2.75 nm shorter than that of the control. These data show that although microfibrils are formed with their main features preserved, the deletion of either one or three domains decreases microfibril periodicity proportionally.

Averages from 458 wild-type periods, 472 ΔH1 periods and 476 WMS periods of the microfibril repeating units were made and analyzed for changes in microfibril structure. In both cases for the mutants, structural features in the shoulder region are lost (Fig. 3c), with the shoulder to the bead peak (seen in the control at ~35 nm) being no longer distinct in WMS and ΔH1 microfibrils. This feature is immediately adjacent to the modeled location of the fibrillin domains through the bead (Fig. 2c). These findings support the location of the TB1 domain in this region and the docking of these domains near the bead region. For both deletions, the perturbation spans a few nanometers, suggesting that the deletions may influence the conformation of neighboring domains.

### LTBP-1 binding is perturbed by a WMS-causing mutation

Given the conformational disruption observed in the mutated microfibrils, we predicted that the WMS deletion of domains TB1–PRR–EGF4

could perturb the interaction of microfibril-binding proteins that bind near to this deletion. To test this, we analyzed the binding of LTBP-1. LTBP-1 is not an integral microfibril component but interacts with the N-terminal region of fibrillin via the hybrid1 domain and adjacent EGF domains[27,46]. To test whether the structural rearrangements observed in the WMS microfibrils could impact on interactions with fibrillin-binding partners, we analyzed the binding of the LTBP-1 C-terminal region to a fibrillin construct containing the WMS deletion. Surface plasmon resonance showed that the affinity of LTBP-1 for fibrillin is decreased with the WMS deletion (Fig. 4a,b). The LTBP-1-binding epitope is two to four domains upstream of the domains deleted in WMS[27]. This further supports longer-range structural rearrangements perturbing a binding site at least 50 Å away when these domains are deleted.

Furthermore, given the domain mapping (Fig. 2c) and analysis of the ΔH1 microfibril structure (Fig. 3c), we hypothesized that LTBP-1 would bind close to the bead region of the microfibril. LTBP-1 is not present (or is only present at very low abundance) in the ciliary zonules and so does not copurify with these microfibrils. Therefore, we bound full-length LTBP-1S to purified fibrillin microfibrils and looked for any increase in mass across the microfibril repeating unit using STEM mass mapping (Fig. 4c). The microfibrils had a mean mass of ~3 MDa per repeat, which is slightly larger than the mass expected for eight fibrillin molecules (~2.7 MDa). This suggests that some microfibril-associated proteins (for example, MAGP1, which is typically present[42,43,47]) might have been copurified. When LTBP-1S was added, there was an increase in mass across the microfibril repeat of 350.2 ± 64.7 kDa, which was localized predominantly to the bead region (Fig. 4c). As the short splice form of LTBP-1 has a molecular weight of 151 kDa, this increase in mass is consistent with two molecules of LTBP-1 binding per microfibril repeat.

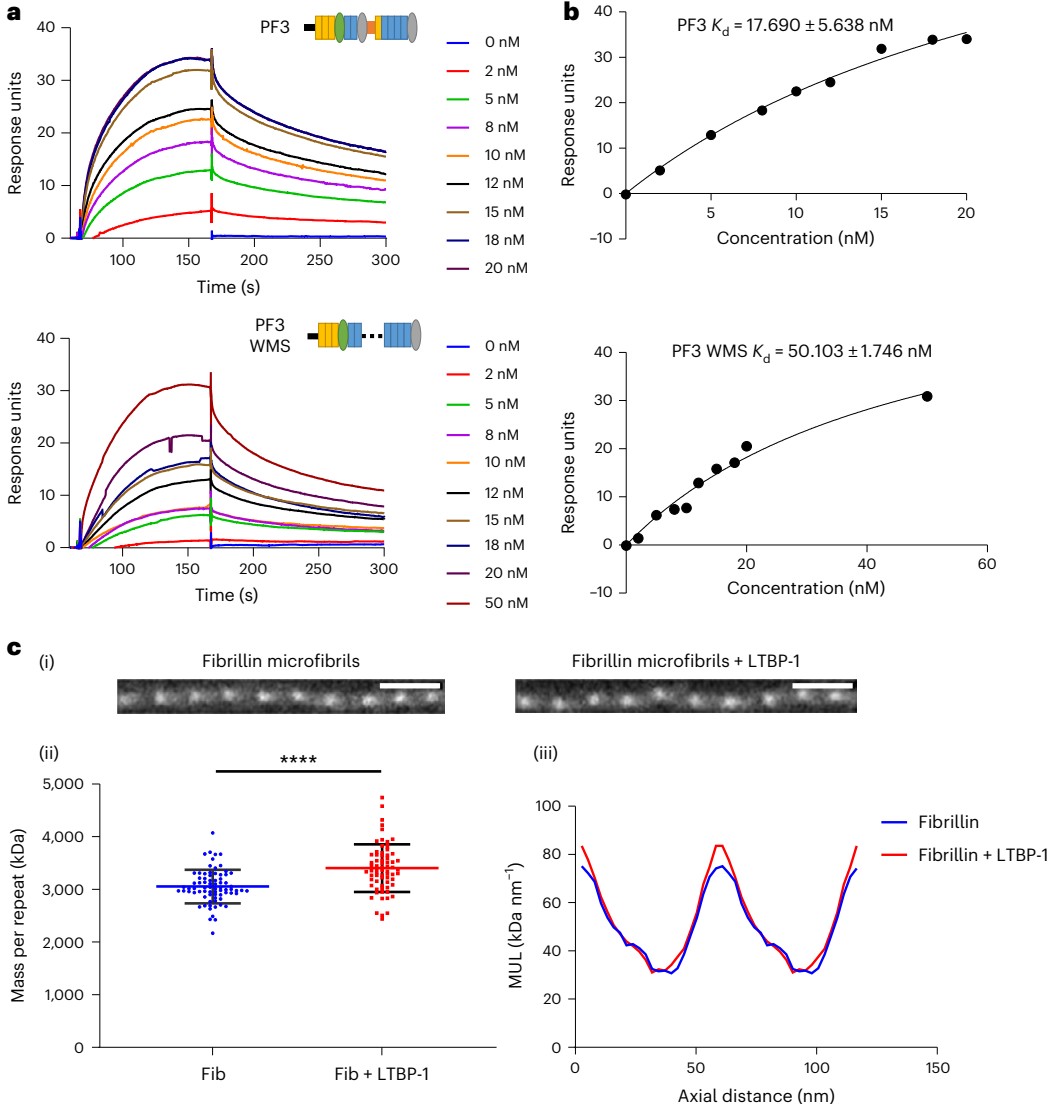

**Fig. 4 | LTBP-1 binds to the bead region of fibrillin microfibrils and is disrupted by a WMS-causing mutation. a**, SPR analysis of LTBP-1 binding to the N-terminal fragment PF3 of fibrillin with and without the WMS deletion. The LTBP-1 C-terminal region was immobilized to the sensor chip using amine coupling, and concentrations of PF3 (0–20 nM) and PF3 WMS (0–50 nM) were flowed over as analytes. Representative sensorgrams show the binding of PF3 or PF3 WMS to LTBP-1. This experiment was repeated three times. **b**, Binding kinetics, as determined using equilibrium analysis for the interaction of PF3 or PF3 WMS with LTBP-1. **c**, A complex of full-length LTBP-1 with purified fibrillin microfibrils was formed and analyzed using STEM mass mapping. (i) STEM images of fibrillin microfibrils with and without LTBP-1. Scale bars, 100 nm. (ii) Scatter plot of the mass per microfibril repeat of fibrillin microfibrils (fib) with and without LTBP-1. The mean mass of a fibrillin repeat was 3,055 ± 36.7 kDa ($n = 75$ from ten images). In complex with LTBP-1, the mass was 3,405 ± 54.43 kDa ($n = 69$ from 11 images). The data represent means ± s.d. Statistical significance was determined by unpaired two-tailed $t$-test (****$P < 0.0001$). (iii) Trace of the mass per unit length (MUL) across the fibrillin repeat showing that there is a gain in mass at the bead region of the microfibril in the presence of LTBP-1. Each trace is an average of 50 periods (ten measurements from each of five images).

## Fibrillin arm region structure

The cryo-EM dataset from the fibrillin microfibrils was next recentered on the arm region and refined with a region-specific 3D mask. Particles that had been aligned to the full fibrillin repeat were recentered to locate the arm region coordinates at the particle center. After 3D classification, the resulting best class was refined using 3D auto-refine with C2 symmetry imposed down the fiber axis, as with the bead reconstruction. The structure of the fibrillin arm region showed eight arms that were continuous from the bead to the interbead region and two shorter densities (Fig. 5a). Domains from TB2 to TB3 were docked into this region, guided by the domains located in the bead region and the antibody mapping of mAb1919 (which recognizes domains cbEGF7–hybrid2). Three distinct bands of higher density were observed in the 2D averages of microfibrils, correlating with the docked locations of

domains TB2, hybrid2 and TB3, which have a larger mass than the EGF/cbEGF domains (Fig. 5c).

## TB4 is localized to the interbead/shoulder junction

Integrin $\alpha_v\beta_3$ binds fibrillin microfibrils through interaction with an RGD (Arg-Gly-Asp) integrin-binding sequence located in TB4[24,48]. To determine where TB4 is located on the mature fibrillin microfibril, purified bovine zonule microfibrils were incubated with the human integrin $\alpha_v\beta_3$ headpiece. Addition of integrin to microfibrils in a 2:1 molar excess led microfibrils to form bundles (Fig. 6a (ii)). These large structures were not seen in the control samples without integrin (Fig. 6a (i)). Averages from 356 control microfibril periods and 391 microfibril periods in the presence of integrin were analyzed for any addition of density along the microfibril repeat. Compared with the control,

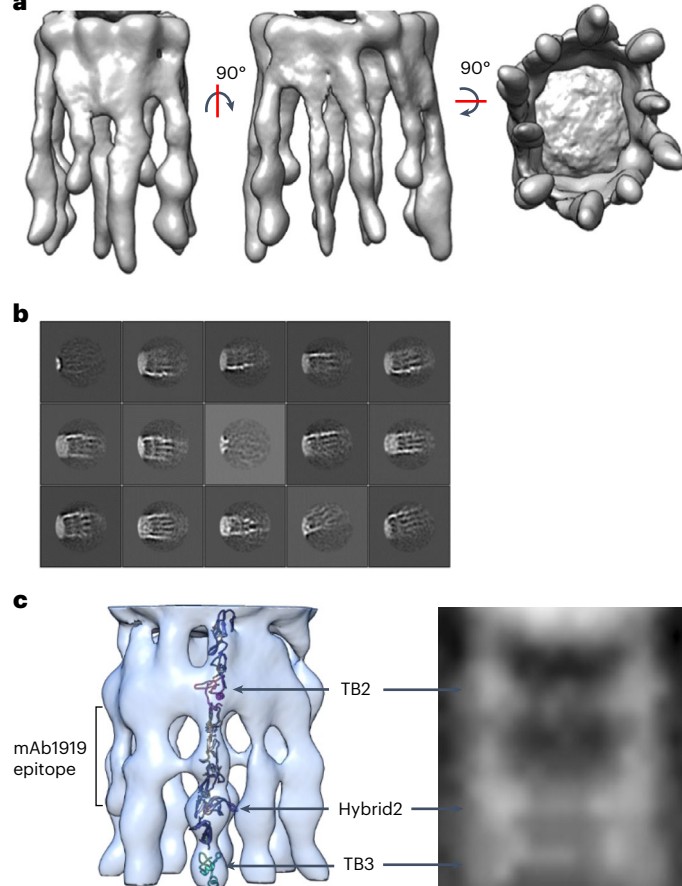

**Fig. 5 | Fibrillin arm region structure.** The cryo-EM classes of the fibrillin microfibrils were recentered and refined to determine the structure of the fibrillin arm region. **a**, Cryo-EM structure of the arm region of fibrillin microfibrils. **b**, 2D class-averaged particles aligned to the fibrillin arm region reconstruction. Box size, 57 nm. **c**, Domains docked into the arm region reconstruction from TB2 to TB3, where the larger TB and hybrid domains correspond to the densities observed in electron microscopy images.

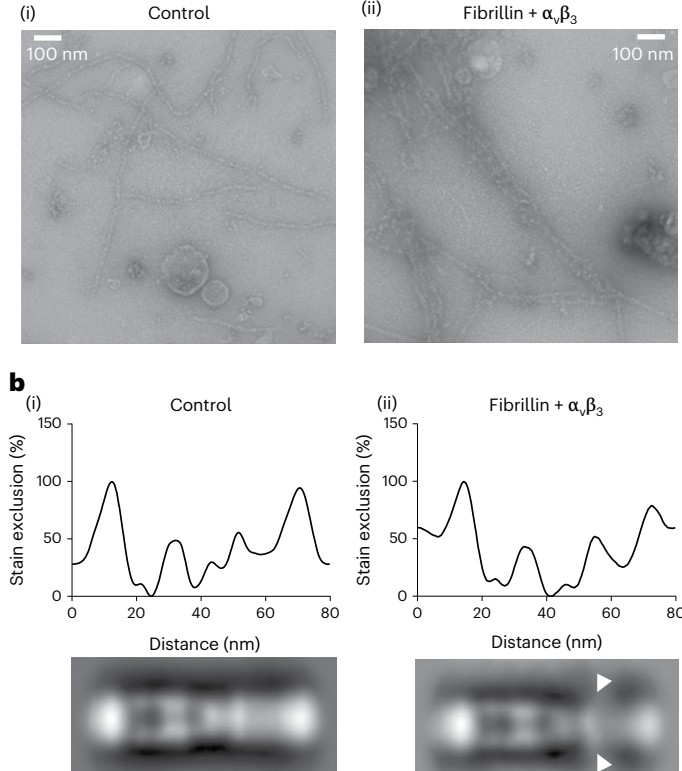

**Fig. 6 | Integrin $\alpha_v\beta_3$ headpiece-binding site. a**, Negative stain images of (i) control microfibrils and (ii) microfibrils in complex with the integrin $\alpha_v\beta_3$ headpiece. **b**, Stain exclusion plots (top) and class average images (bottom) of (i) 356 control microfibril periods and (ii) 391 microfibril periods complexed with the integrin $\alpha_v\beta_3$ headpiece. The arrowheads indicate a region at the interbead/shoulder junction with less well-defined structure and diffuse stain exclusion in the presence of the integrin $\alpha_v\beta_3$ headpiece. The aligned and averaged microfibril periods from two biological repeats are shown.

the averaged image of the microfibril–integrin complex (Fig. 6b (ii)) had an increase in stain exclusion in the interbead region next to the shoulder region. This region had a less well-defined structure with diffuse stain exclusion, observed as peak broadening in the stain exclusion plot (Fig. 6b). Due to the multimeric nature of the microfibril, there are multiple copies of the fibrillin molecule and thus the TB4 domain in each microfibril period. The diffuse staining is probably due to variable numbers of integrin $\alpha_v\beta_3$ molecules binding on different microfibril periods, and there could be potential variability in the presentation of the RGD-containing loop on the microfibril, so the integrin could be bound in different orientations. The extra density in the interbead–shoulder junction region of the microfibril suggests that the TB4 domain is in this region.

## Discussion

Here we present the cryo-EM structure of a native fibrillin microfibril from mammalian tissue. Extracellular matrix fibrils are resistant to structural biology approaches due to their complexity. Here we utilized microfibrils from ciliary zonules as they are elastin free (forming the major zonular component) and can be purified without denaturation or the use of proteases. Using ultrastructural analysis of microfibrils from different tissues, we found that the microfibril backbone is consistent between tissues with only minor differences[18]. This backbone should serve as a conserved deposition scaffold for tropoelastin and

adaptor molecules, such as LTBPs or fibulins, where the amounts of microfibril-binding proteins may differ in a tissue-specific manner.

The microfibril was subdivided into different regions for image analysis, to overcome flexibility along the microfibril axis. The highest resolution was achieved for the bead region, with subnanometer resolution. Guided by antibody mapping and the location of domain deletions on the microfibril, we docked an N-terminal region of fibrillin composed of domains TB1–cbEGF6 into the center of the bead. Peptides from this region are rarely detected in mass spectrometry analysis of microfibrils digested under nondenaturing conditions, which suggests that they are inaccessible to proteolytic enzymes and buried within the microfibril structure. When microfibrils are extracted under denaturing conditions, this region is present[43], which indicates that it may be buried and inaccessible under native conditions.

We considered alternative arrangements for the path of the molecule through the bead. The density at the edge of the bead connects into the arm region unambiguously, which makes us confident in our assignment of domains cbEGF5 and cbEGF6. However, it is possible that upstream of these domains the molecule deviates around the bead rather than through it. In this scenario, the C-terminal domains could form part of the central core of the bead. The C-terminal half of fibrillin has been shown to independently form multimers with a beaded structure and may initiate microfibril assembly by binding the N-terminal region[16].

Microfibril assembly proceeds via head-to-tail assembly of fibrillin molecules[14], where interaction between the terminal

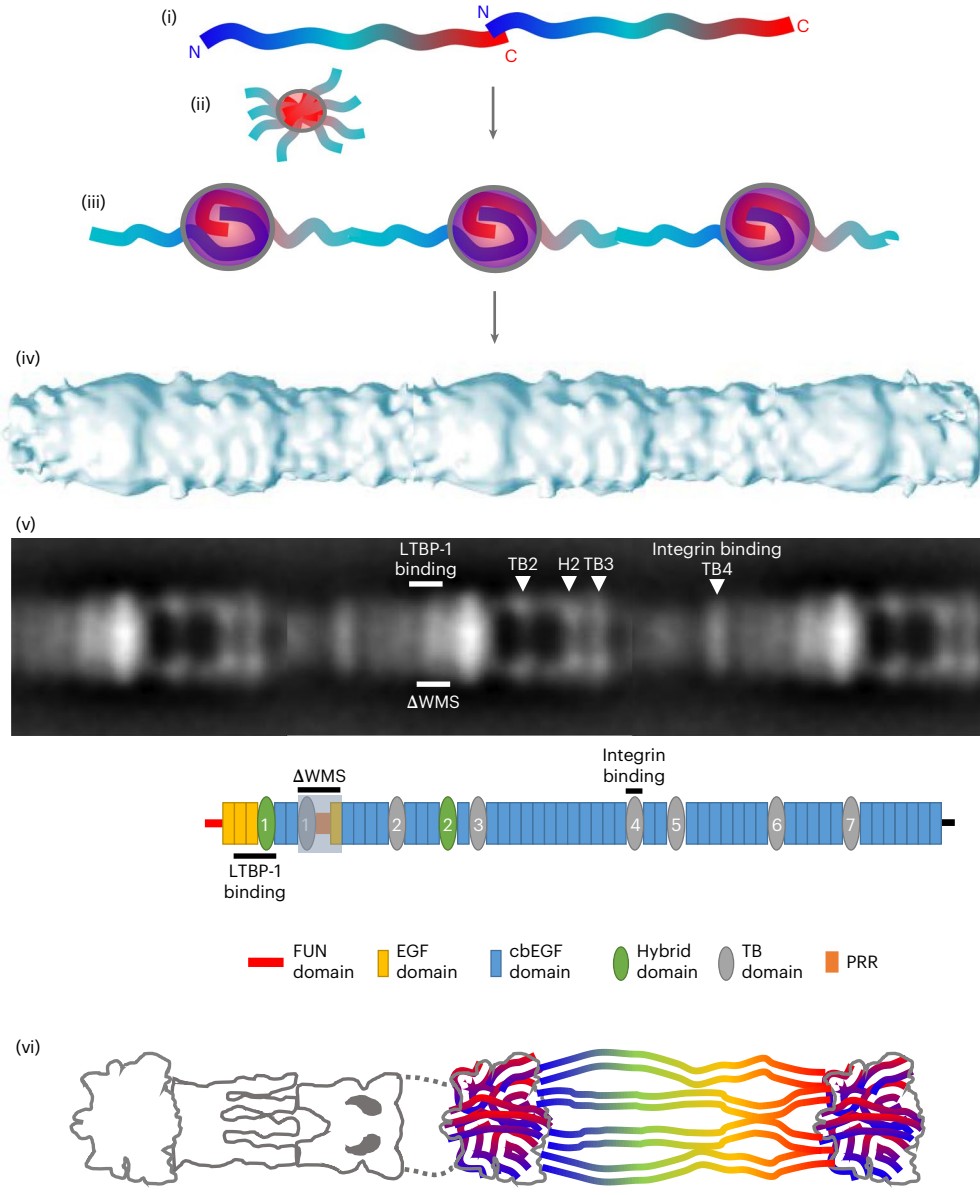

**Fig. 7 | Fibrillin assembly and organization.** (i) Published data support an initial head-to-tail assembly of fibrillin molecules mediated via an interaction between domains FUN–EGF1 and cbEGF41–43 (refs. 14,49). Subsequent assembly steps result in a mature beaded microfibril with N- and C-terminal antibody epitopes either side of the bead region, pseudo eightfold symmetry and 57 nm periodicity. (ii) The C-terminal region of fibrillin is likely to facilitate assembly as the C-terminal half of fibrillin-1 can independently multimerize into bead-like structures[16]. (iii) The bead has a mass of ~1.1 MDa, as determined by STEM mass mapping and the cryo-EM density map, so if eight pairs of fibrillin molecules overlap in a head-to-tail manner within the bead, the mass would relate to >20

fibrillin domains present for each overlapping pair of molecules. This supports more extensive secondary interactions[15,50] and packing of terminal regions within the bead. (iv) The mature microfibril has a 3D cylindrical structure with pronounced beads on a string when viewed in projection. (v) Regions mapped in this study are shown on a fibrillin microfibril negative stain class average, with the same regions shown below on a schematic of the fibrillin molecule. (vi) Together our data show that the bead is dense and interwoven, supporting condensation of N- and C-terminal regions within and the location of other regions of the fibrillin molecule within the arm, interbead and shoulder regions.

domains is supported by binding data showing that a C-terminal region (cbEGF41–43) interacts with an N-terminal region (the fibrillin-unique N-terminal region (FUN)–EGF1)[49]. However, the volume and mass of the bead supports more extensive packing of N- and C-terminal domains within the bead region (Fig. 7). It is likely that after initial head-to-tail interaction further domains are interwoven within the bead, which would increase the density and mass of this region (consistent with STEM mass and atomic force microscopy height data) compared with other regions of the microfibril[38]. Further overlap or interactions between the N- and C-terminal regions is also supported by binding data that show interactions between domains

upstream and downstream of the N- and C-terminal regions, respectively (specifically, interaction between two N-terminal regions upstream and downstream of TB1 and interaction between the N-terminal region and a more C-terminal region upstream of TB6 (refs. 15,50)). Furthermore, it is supported by the positioning of a tissue transglutaminase crosslink between the N-terminal cbEGF5 domain and C-terminal TB7 domain[51], although this crosslink may be between microfibrils rather than within one.

The shoulder region was difficult to resolve using cryo-EM with single-particle averaging, suggesting that this region has some conformational heterogeneity. The perturbation to the shoulder in ΔH1

microfibrils suggests that the FUN region is nearby and probably resides in the shoulder region. NMR data have shown that the FUN region is flexible and likely to be conformationally heterogeneous[49]. The shoulder region of the microfibril is also likely to be very compositionally heterogeneous, as the N-terminal region binds to the majority of microfibril-binding proteins (as reviewed in ref. 2).

To analyze the structural consequences of domain deletions on the microfibril and locate specific important domains, microfibrils from mice with deletions of the hybrid1 domain and a WMS-causing deletion (lacking TB1–PRR–EGF4 domains) were analyzed. Microfibrils containing either deletion had decreased periodicity, concomitant with the number of domains deleted, and showed disorganization of the shoulder region. Our docking of the TB1–TB2 region supports the location of the first hybrid and TB1–PRR–EGF4 domains within or adjacent to the shoulder region. These structural perturbations suggest that the mutated microfibrils may be less stable than the wild type. Indeed, elastic fibers in the skin of patients with WMS and mice carrying the WMS deletion have a moth-eaten appearance with abnormal aggregates of microfibrils[36].

The disruption of the microfibril structure was similar for both deletions, suggesting that both deletions may structurally perturb the conformation of neighboring domains. Therefore, we hypothesized that the WMS deletion may disrupt binding to the hybrid1 domain, where latent TGFβ is known to bind via LTBP-1 (ref. 27). Having confirmed that LTBP-1 binds the bead region of the microfibril, we also showed that a fibrillin-1 region containing the WMS deletion had decreased affinity for LTBP-1. However, TGFβ signaling does not appear to be disrupted in human WMS tissues or in WMS mouse models[36,52], suggesting that the perturbation in binding does not directly contribute to the pathomechanism of disease.

The results from our study also contribute substantially to the current understanding of how structural alterations in fibrillin microfibril ultrastructure caused by fibrillin mutations lead to growth factor dysregulation. Currently, the bead region is seen as a critical microenvironment for growth factor sequestration and controlled activation[53,54]. Any perturbation in this sensitive microenvironment leads to dysregulated growth and differentiation processes resulting in opposing clinical features (for example, long bone overgrowth versus undergrowth or low muscle tone versus hypermuscularity) of the fibrillinopathies. We have shown that two LTBP-1 molecules can bind per bead unit. Given the length of LTBP-1 and the proximity of the hybrid1 and FUN domains, it is conceivable that binding of LTBP-1 may also restrict BMP binding or its activation. Furthermore, we have shown that the introduction of the WMS deletion in the fibrillin molecule perturbs a firm LTBP-1 interaction with fibrillin. The structural rearrangement due to the WMS deletion mutation may prevent proper activation of sequestered BMPs from the microfibril scaffold, as seen by us and others in WMS mouse models[52,55].

When reconstituted in vitro, two LTBP-1 molecules per repeat were able to bind to fibrillin microfibrils. Considering that there are eight fibrillin molecules per microfibril period[44], this suggests that the other sites are either already occupied or there is steric hindrance preventing more LTBP-1 molecules from binding. LTBP-1 is a large elongated molecule composed of more than 20 domains that can extend to ~40 nm in length[56]. The mass increase upon LTBP-1-binding appears localized across the bead, which suggests that interactions can occur from the shoulder to the bead region, consistent with the elongated nature and length of LTBP-1. Therefore, there could be steric effects preventing further LTBP-1 molecules from binding. Furthermore, fibulins-2, -4 and -5 also bind to the first hybrid domain[27], which if copurified with microfibrils could compete for the LTBP-1 binding site, although mass spectrometry data indicate that the fibulins are not major components in the ciliary zonule, and fibulin-2 is found in the vitreous[43]. However, a number of proteomics studies have shown that LTBP-2 is associated with microfibrils in ciliary zonules[41–43], which were probably copurified

in our microfibril preparations and may compete with LTBP-1 for the available epitopes.

Indeed, in the reconstruction of the arm and bead regions, there are ten arms that extend out from the bead. Eight of these are of similar size, corresponding to the eight fibrillin molecules, but two are shorter and terminate at the edge of the bead rather than continuing into the interbead (Extended Data Fig. 3). It is not clear whether the presence of these two truncated arms is an artifact of the 3D reconstruction due to the flexibility of some of the arms or distortion of the structure by interaction with the air–water interface, or whether the two smaller arms correspond to two microfibril-binding proteins present in the microfibril structure. Indeed, STEM mass mapping performed on these microfibrils gave an experimental mass of ~3 MDa, which is larger than was previously observed for microfibrils from canine ciliary zonules (2.55 MDa)[38]. Moreover, the expected mass of eight fibrillin molecules would be around 2.7 MDa, suggesting that microfibril-associated proteins may have copurified in this preparation. Mass spectrometry data of ciliary zonule microfibrils after gel filtration showed a number of LTBP-2 peptides (unpublished data), indicating that LTBP-2 copurifies with these microfibrils, and LTBP-2 constitutes ~7% of proteins (fibrillin-1 ~75%) in the bovine zonule[43]. The LTBPs are members of the fibrillin superfamily and have a similar domain structure to fibrillin, so they would be expected to have a similar diameter and shape to fibrillin molecules; therefore, it is possible that microfibril-binding proteins have been observed in the 3D reconstruction of the microfibril.

Our protease-free purification may lead to better preservation of the microfibril ultrastructure. It has been shown that microfibril isolation by enzymatic digestion destroys BMP-binding epitopes[20]. Therefore, our method allows us to investigate growth factor binding epitopes within the ultrastructure of the microfibril. Furthermore, binding of the integrin $\alpha_v\beta_3$ headpiece to the microfibril shows the positioning of the RGD motif-containing TB4 domain, highlighting the presence of other functional epitopes on the microfibril.

## Online content

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

## Methods

### Ethics

Isolation of fibrillin microfibrils from murine tissues was carried out in strict accordance with the German federal law on animal welfare, and the protocols were approved by the Landesamt für Natur, Umwelt und Verbraucherschutz Nordrhein-Westfalen for breeding (permit number 84-02.04.2014.A397) and euthanasia (permit number 84-02.05.40.14.115).

### Fibrillin microfibril extraction from bovine ciliary zonule

For cryo-EM analysis, ciliary zonules were extracted from dissected bovine ciliary zonule tissue and were disrupted using sonication in 20 mM Tris-HCl, 400 mM NaCl and 2 mM $CaCl_2$ (pH 7.4) with two 10-s pulses and cooled on ice for 30 s between pulses. Samples were centrifuged and the supernatant was separated using size exclusion chromatography with a 5 ml CL-2B resin column. The column was equilibrated in either the sonication buffer or a physiological salt buffer for cryogenic transmission electron microscopy studies (20 mM Tris-HCl, 150 mM NaCl and 2 mM $CaCl_2$ (pH 7.4)). The void volume containing fibrillin microfibrils was saved for further electron microscopy analysis. Each purification utilized the ciliary zonules from a single bovine eye. Separate purifications were performed for the STEM mass mapping and cryo-EM experiments and multiple purifications were used for optimization of cryo-EM grid preparation and data collection.

### Fibrillin microfibril extraction from ΔH1 and WMS mice

Wild-type C57BL/6 mice or mice homozygous for either deletion of the first hybrid domain (ΔH1)[35] or a deletion of three domains (TB1–PRR–EGF4) that causes WMS[36] were maintained at a temperature of 20–24 °C under 50–70% relative humidity with a light cycle of 12 h day and 12 h night. A 1 cm$^2$ region was dissected from the back skin of 6-week-old males and digested overnight at 4 °C in digestion buffer (0.1 mg ml$^{-1}$ Collagenase Type I-a (Sigma–Aldrich) in 20 mM Tris-HCl, 400 mM NaCl and 2 mM $CaCl_2$ (pH 7.4)). Samples were centrifuged and separated using size exclusion chromatography as described above. The purification and imaging were repeated at least twice on different skin samples.

### Negative stain electron microscopy

Microfibrils extracted from the skin of mice with the mutations and control C57BL/6 mice were adsorbed onto glow-discharged carbon-coated copper grids (Agar Scientific) and stained with 2% uranyl acetate. Grids were imaged at a magnification of 23,000× on a Tecnai G2 Polara TEM (FEI) equipped with a K2 Summit direct detector (Gatan) operating at an accelerating voltage of 300 kV. Images were collected over a 1 s exposure time in linear mode and sampled at 1.4 Å pixel$^{-1}$. The microfibril repeat was manually picked in RELION-2.1 (ref. 57) and subjected to 2D class averaging.

To determine the period length of the ΔH1 and WMS mutated microfibrils, the periodicities of 100 microfibril repeats were measured using ImageJ[58]. Microfibrils were extracted from images and straightened using the straighten tool before measuring the repeat length.

### Cryo-EM sample preparation and data collection

Microfibril samples were adsorbed onto glow-discharged holey carbon Quantifoil R1.2/1.3 grids before being blotted and plunge frozen in liquid ethane using a Vitrobot Mark IV (FEI). Microfibrils were imaged using automated data collection in EPU on a Titan Krios electron microscope (FEI) operating at an accelerating voltage of 300 kV. A total of 1,310 videos comprising 20 frames with a 20-s exposure time and a total dose of 66 e Å$^{-2}$ were collected on a K2 Summit direct detector (Gatan) in counting mode at a nominal magnification of 64,000, which gave a pixel size of 2.2 Å. Initial imaging of microfibrils showed that they had a preferred orientation on the grid, so during data collection the stage was tilted to 45°.

### Symmetry analysis of 3D maps

Orientation correlation plots were made by rotating the 3D maps refined in RELION-2.1 (pre-postprocessing) about the imposed C2 symmetry axis or the putative symmetry axis for the C1 map (Extended Data Fig. 1a). Each map was rotated over a series of angles, from 0 to 360° in increments of 5°, and correlated back to the original unrotated map. Normalized correlation coefficients were calculated in a map region defined by a binary mask derived from the mask used for the refinements. This original mask was symmetrized by rotating sequentially over 5° steps (range 0–360°) to create a series of masks and then taking their sum.

Where two independent half-maps were used, one was kept constant for comparison and the other was rotated around the symmetry axis (Extended Data Fig. 1a). The program SPIDER (version UNIX 26.04) was used for the map rotations and correlation coefficient calculations[59].

### Single-particle data processing bead region reconstruction

Videos were motion corrected and dose weighted using MotionCor2 (ref. 60). Corrected images were imported into Warp[61] where 27,737 particles were picked using a Warp box net that had previously been trained on manually picked particles of the fibrillin microfibrils. Patch-based contrast transfer function (CTF) estimation was used to calculate local CTF values for the particles. Particle stacks were imported into cryoSPARC[62] and used in a homogenous refinement using a negative stain reconstruction of the microfibril repeat as an initial model[37]. The resulting structure from cryoSPARC was then further refined in RELION-2.1 (ref. 57). Particles that had been aligned to the full fibrillin repeat were recentered so that the bead or arm region coordinates were then at the particle center using a custom Python script. For the bead region reconstruction, the shifted particle coordinates were then extracted and were 2D classified without alignment to remove the bad particles; 13,184 particles from good classes were selected for 3D classification. The 3D classification of the shifted particles was performed with a restricted angular search using the command --sigma_ang 5. The resulting best class was then further refined using 3D auto-refine with C2 symmetry imposed down the fiber axis. To remove any remaining bad particles, a nonaligned 2D classification was performed and 7,139 particles were used in a final refinement. After postprocessing in RELION, the final structure had a resolution of 9.7 Å (Extended Data Fig. 1b). Local resolution estimation using the two half-maps from the final bead refinement was performed in cryoSPARC (Extended Data Fig. 1d).

For the arm region reconstruction, the shifted particle coordinates were also extracted and 2D classified without alignment to remove bad particles; 4,957 good particles were selected for 3D classification. The 3D classification of the shifted particles was then performed with a restricted angular search using the command --sigma_ang 5. The resulting best class was refined using 3D auto-refine with C2 symmetry imposed down the fiber axis, as with the bead reconstruction. After postprocessing in RELION, the final arm structure had a resolution of 18.3 Å (Extended Data Fig. 4). The electron microscopy data have been deposited to the Electron Microscopy Data Bank (EMDB) with accession codes EMD-13984 for the bead model and EMD-13986 for the arm region.

### Refining antibody epitopes with recombinant fibrillin fragments

Recombinant human fibrillin-1 fragments were expressed and purified as previously described[19,63]. Proteins were detected by western blotting with the antibodies mab2502 (clone 26) and mab1919 (clone 11C1.3) from Sigma–Aldrich, using a dilution of 1:1,000.

### LTBP-1 binding and STEM mass mapping

Full-length human LTBP-1 was purified as previously described[56] and incubated with bovine fibrillin microfibrils at a 2:1 molar ratio for 4 h at 4 °C. The complex was adhered for 60 s to 400-mesh carbon-coated grids then washed with Milli-Q water three times and dried. The sample was visualized in STEM mode with a Fischione high-angle annular

dark-field detector on an FEI Tecnai 12 Twin TEM at 34,000× magnification. A camera length of 350 cm was used to give an angular collection range of 20–100 mrad. Tobacco mosaic virus was used as a calibration standard for mass per unit length. The electron dose was kept sufficiently low (<300 e nm$^{-1}$) to produce negligible mass loss. Mass per unit length measurements and axial mass distributions were measured from STEM annular dark-field images using the Semper6 image analysis software (Synoptics).

## Fibrillin–LTBP-1 binding by surface plasmon resonance

The C-terminal region of human LTBP-1 was purified as previously described[56] and immobilized by amine coupling onto a CM5 sensor chip at 2.5 µg ml$^{-1}$ in 50 mM sodium acetate buffer (pH 3.0) in a Biacore T200 biosensor; typically, 200 response units were immobilized. Recombinant human fibrillin-1 fragments PF3 and PF3 WMS (0–50 nM) were injected onto the sensor chip at a flow rate of 50 µl min$^{-1}$ in 10 mM HEPES, 150 mM NaCl, 1 mM CaCl$_2$ and 0.05% Tween-20 for 100 s and then allowed to dissociate for 180 s. Regeneration was performed by injection of 10 mM glycine (pH 2) for 30 s at a flow rate of 30 µl min$^{-1}$ followed by a stabilization period of 60 s. $K_d$ values were calculated using equilibrium analysis: equilibrium response was plotted against concentration and nonlinear regression was used to calculate $K_d$ values using the equation for one-site binding. Each assay was performed at least twice.

## Complexing integrin α$_v$β$_3$ with fibrillin microfibrils

Integrin-αv Headpiece M400GC and pcDNA3.1-β3 were gifts from T. Springer (plasmids 159637 and 27289; Addgene)[64,65]. The human integrin α$_v$ headpiece (residues 1–594) and the β$_3$ headpiece (residues 1–472) with a mutation causing the amino acid alteration p.Gln267Cys were subcloned and ligated into the pBudCE4.1 mammalian expression vector. The α$_v$ headpiece contains a C-terminal acid coiled coil and a Twin-Strep tag, whereas the human β$_3$ headpiece has a C-terminal base coiled coil and a His tag.

The recombinant construct was transiently transfected into the Expi293 mammalian expression system (Thermo Fisher Scientific) according to the user guide. The integrin α$_v$β$_3$ headpiece was then affinity purified using Strep-Tactin XT 4Flow resin (IBA Lifesciences), followed by size exclusion chromatography on a Superdex 200 increase column (Cytiva) equilibrated in buffer containing 20 mM HEPES and 150 mM NaCl (pH 7.5).

Bovine ciliary zonule microfibrils and integrin α$_v$β$_3$ were incubated together in an approximate 2:1 integrin-to-microfibril molar ratio in 20 mM Tris-HCl, 150 mM NaCl, 500 µM Mn$^{2+}$ and 500 µM Ca$^{2+}$ (pH 7.4) for 1 h at 4 °C. Microfibrils with and without integrin α$_v$β$_3$ were then imaged using negative stain electron microscopy as described above.

### Reporting summary

Further information on research design is available in the Nature Portfolio Reporting Summary linked to this article.

## Data availability

The cryo-EM data have been deposited to the Electron Microscopy Data Bank with accession codes EMD-13984 (for the bead model) and EMD-13986 (for the arm region). Source data are provided with this paper.

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

## Acknowledgements

We thank staff at the Electron Microscopy Core Facility (RRID:SCR_021147; Faculty of Biology, Medicine and Health, University of Manchester) for assistance and the Biotechnology and Biological Sciences Research Council (BB/T017643/1) for funding toward the Glacios cryo-EM used for screening. The Wellcome Trust Centre for Cell-Matrix Research is supported by funding from the Wellcome Trust (203128/Z/16/Z). A.R.F.G. was supported by Biotechnology and Biological Sciences Research Council funding (BB/N015398/1 to C.B., A.M.R. and M.J.S. and BB/S015779/1 to C.B.). We acknowledge the UK national Electron Bio-Imaging Centre (eBIC) for access to cryo-EM facilities (EM16619-5). Data were collected at the Astbury Biostructure Laboratory on FEI Titan Krios microscopes, which were funded by the University of Leeds and Wellcome Trust (108466/Z/15/Z). Funding for this study was also provided by Deutsche Forschungsgemeinschaft project numbers 73111208 (SFB 829/B12), 384170921 FOR2722/C2 and 397484323 TRR259/B09 to G.S.

## Author contributions

A.R.F.G. performed all of the experiments and data analysis, unless stated otherwise. R.D. and X.Z. performed the integrin–microfibril binding analysis. J.T. performed the fibrillin epitope mapping. D.F.H. analyzed the STEM mass mapping data. C.S.A. and G.S. provided the ΔWMS and ΔH1 samples. M.J.S., A.M.R. and C.B. designed and supervised the research. A.R.F.G. and C.B. interpreted the data and wrote the manuscript with input from A.M.R., M.J.S. and G.S.

## Competing interests

The authors declare no competing interests.

## Additional information

**Extended data** is available for this paper at https://doi.org/10.1038/s41594-023-00950-8.

**Correspondence and requests for materials** should be addressed to Clair Baldock.

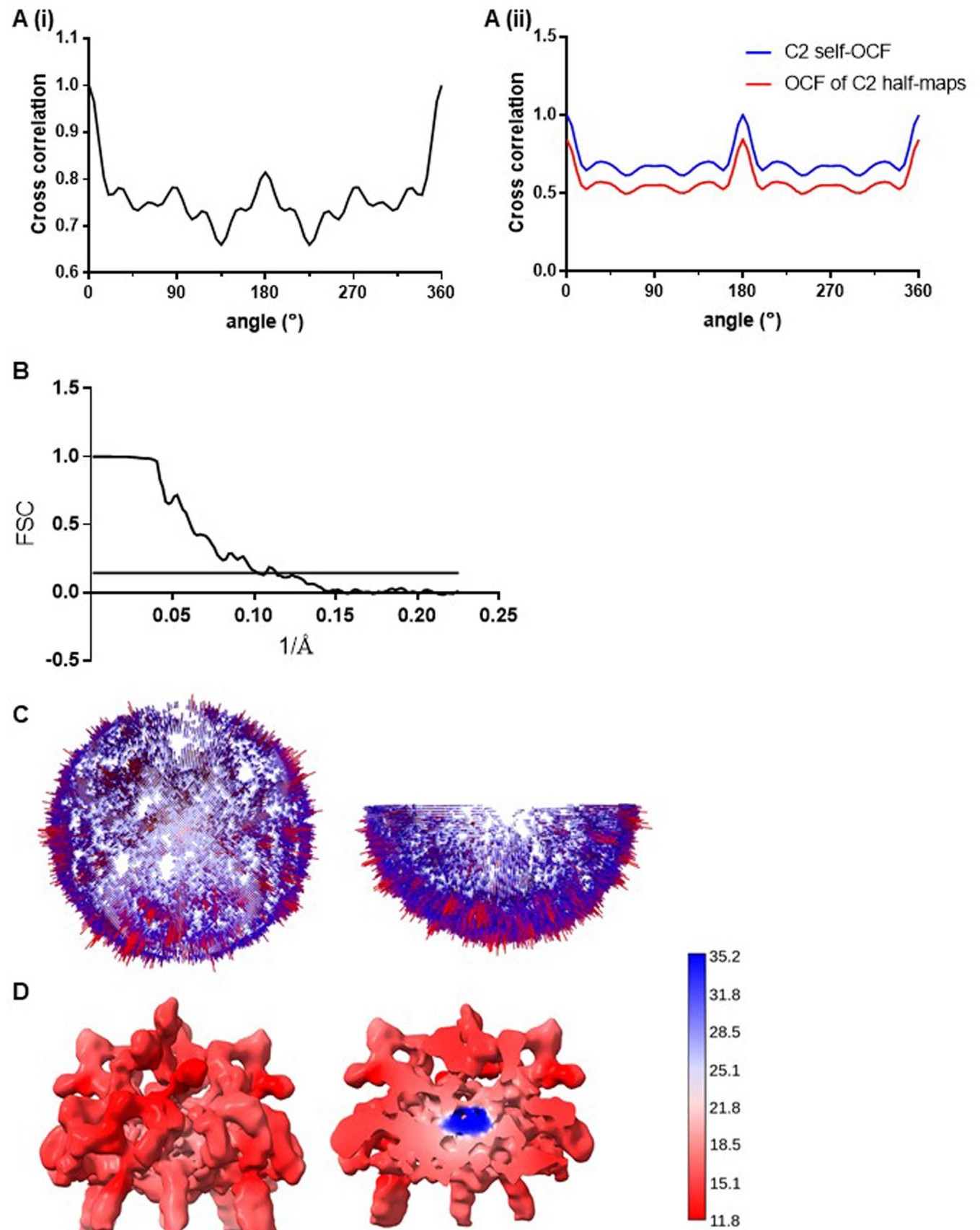

**Extended Data Fig. 1 | Fibrillin bead region cryoEM data. a)** Orientation correlation plots of the microfibril bead region processed with no symmetry (i) or with C2 symmetry imposed during the refinement (ii). (**b**) Fourier shell correlation (FSC) of the fibrillin microfibril bead structure. Resolution was calculated from the correlation between two independently refined halves of the data and is 9.7 Å resolution at 0.143 criterion. **c**) A 3D representation of the angular distribution of particles used in the bead structure reconstruction. (D) The final bead postprocessed reconstruction was surface coloured in chimeraX by local resolution estimation calculated in cryoSPARC.

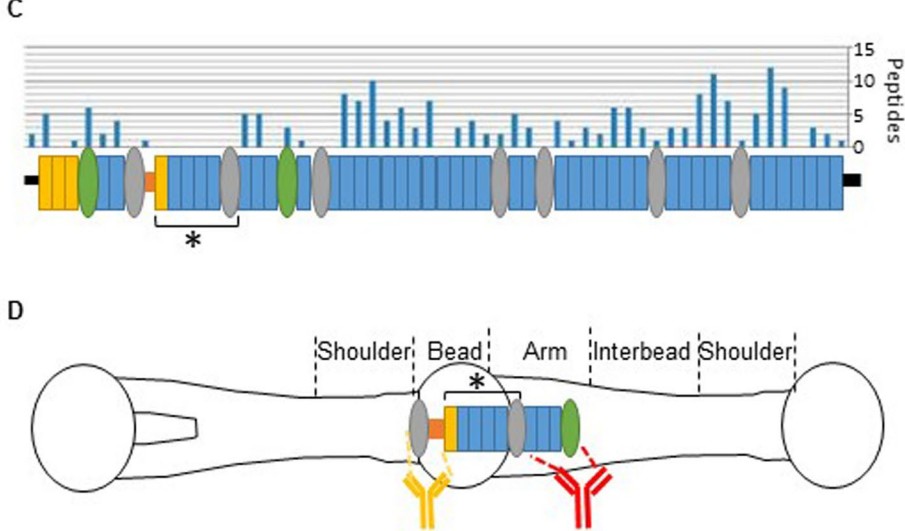

## A (i)

Fibrillin-1 — 45-450 — Mab2502

| | | Mab2502 |
|---|---|---|
| PF2 (43 kDa) | | √ |
| PF1 (48 kDa) | | √ |
| Ex3-11 (44 kDa) | | √ |
| Ex4-11 (40 kDa) | | √ |
| PF4 (30 kDa) | | X |

330-450
TB1-PRR
Refined epitope

## A (ii)  Mab2502

Ex3-11 (R NR), Ex4-11 (R NR), PF1 (R NR), PF2 (R NR), PF4 (R NR)

250, 150, 100, 75, 50, 37, 25, 20

250kDa, 150kDa, 100kDa, 75kDa, 50kDa, 37kDa, 25kDa, 20kDa

## B (i)

Fibrillin-1 — 451-909 — Mab1919

| | | Mab1919 |
|---|---|---|
| N-Ter (165 kDa) | | √ |
| PF3 (80 kDa) | | X |
| PF2 (43 kDa) | | X |
| PF5 (37 kDa) | | √ |

723-909
cbEGF7-Hyb2
Refined epitope

## B (ii)  Mab1919

PF5 (R NR), PF2 (R NR), PF3 (R NR), N-Ter (R NR), PF1 (R NR)

250kDa, 150kDa, 100kDa, 75kDa, 50kDa, 37kDa, 25kDa, 20kDa

Legend:
- EGF domain
- cbEGF domain
- hybrid domain
- TB domain
- Proline-rich region

## C

Peptides (scale 0–15)

*

## D

Shoulder | Bead | Arm | Interbead | Shoulder

*

Mab2502/69        Mab1919/11C1.3

**Extended Data Fig. 2 | See next page for caption.**

**Extended Data Fig. 2 | Refining epitope labelling of fibrillin recombinant fragment.** Mab2502 (clone 26 (epitope within residues 45–450 from epitope mapping[13])) and Mab1919 (clone 11C1.3 (epitope within residues 451–909 as defined by manufacturer)) were used to probe overlapping recombinant fibrillin fragments to more accurately define their epitopes in fibrillin-1. (Ai and Bi) schematic diagrams of recombinant fragments N-Ter, PF1, PF2, PF3, PF4, PF5, Ex3-11 and Ex4-11. Recombinant fibrillin-1 fragments after separation by SDS-PAGE in the presence (R) or absence (NR) of a reducing agent followed by western blotting with either (Aii) Mab2502 or (Bii) Mab1919. These blots were repeated at least 3 times. The antibody epitopes for mab2502 (45–450[13]) and mab1919 (451–909)

are narrowed down to TB1-PRR (residues 330–450) and cbEGF7-Hyb2 (residues 723-909) respectively, highlighted in red. (C) Number of peptides identified by LC-MS/MS from each domain of fibrillin, where a protease resistant region is located between TB1 to TB2 (TB domains are numbered) and indicated by an asterisk (redrawn from[41]). (D) Diagram of the fibrillin microfibril repeating unit with the binding sites of Mab2502 and Mab1919 in adult human ciliary zonule microfibrils (as determined in[38]) and putative location of the protease resistant region identified in[41]. Panel c adapted from ref. 41 under a Creative Commons licence CC BY 4.0.

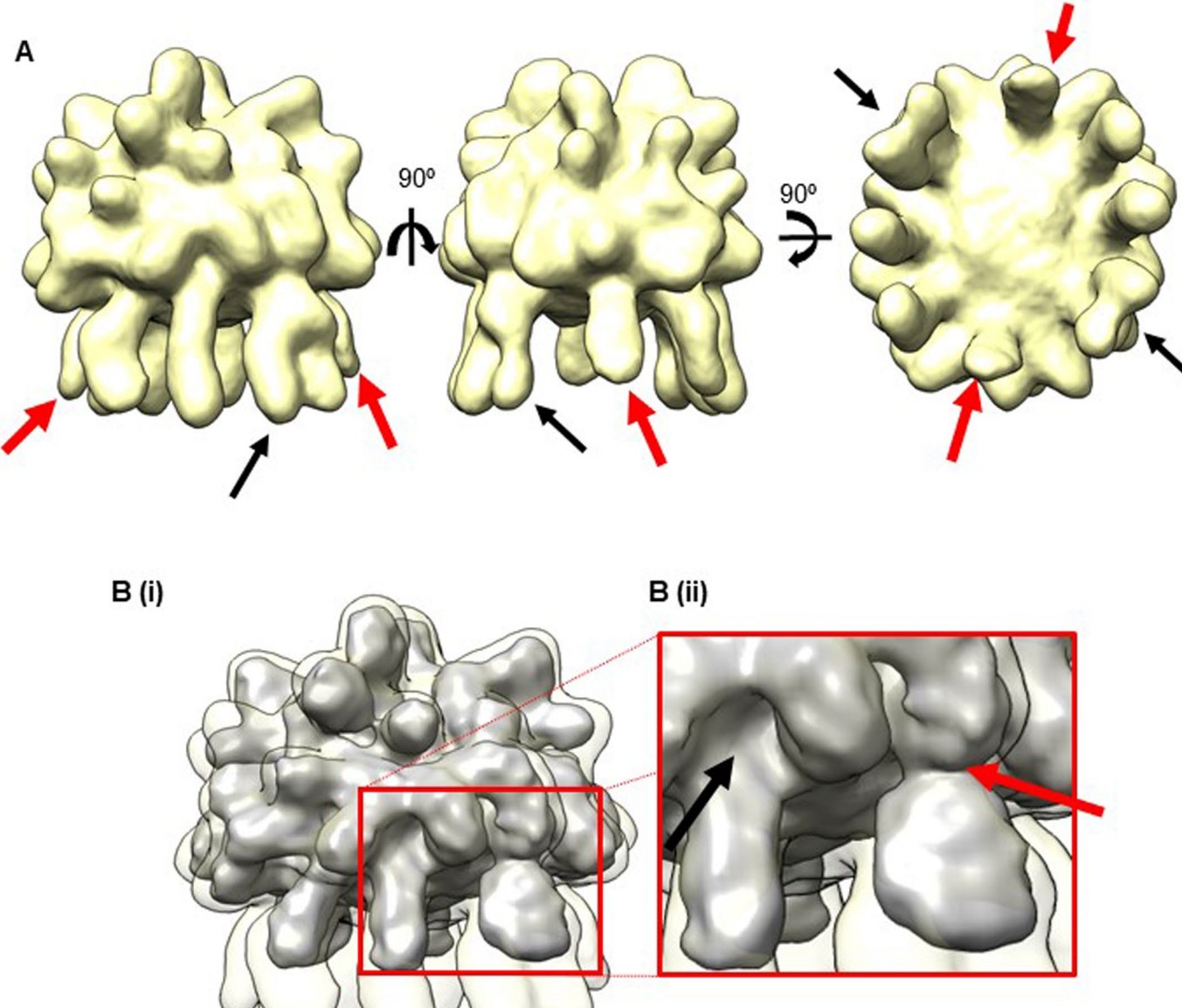

**Extended Data Fig. 3 | Fibrillin arm region reconstruction. a)** 3D reconstruction of the bead region shown in three orthogonal orientations. There are 8 arms which are visible emerging from the bead structure. Highlighted with black arrows are two arms which are wider. The red arrows highlight two arm regions which are masked out in the final bead reconstruction. **b)** The reconstruction in (A) is shown overlaid on the final bead reconstruction shown in grey. Panel (Bii) shows a magnified view of the arm regions connecting to the bead region. The red arrow shows how the wider arms connects to the outside of the bead, in contrast the arm highlighted by the black arrow inserts into the centre of the bead.

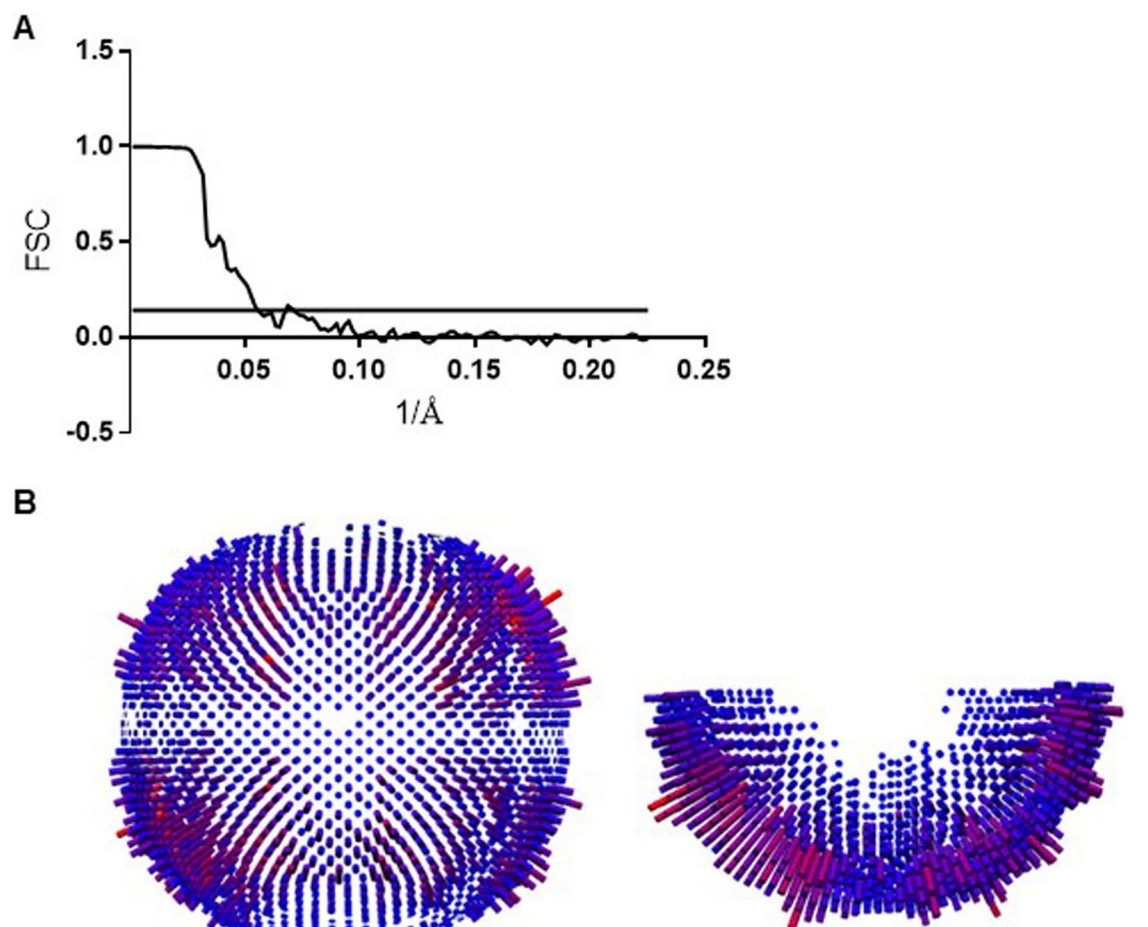

**Extended Data Fig. 4 | Fibrillin arm region cryoEM data. a**) A Graph of the Fourier shell correlation (FSC) of the fibrillin microfibril arm structure. Resolution was calculated from the correlation between two independently refined halves of the data and is 18.3 Å resolution at 0.143 criterion. **b**) A 3D representation of the angular distribution of particles used in the arm structure reconstruction.

# Reporting Summary

## Statistics

For all statistical analyses, confirm that the following items are present in the figure legend, table legend, main text, or Methods section.

| n/a | Confirmed | |
|---|---|---|
| ☐ | ☒ | The exact sample size (*n*) for each experimental group/condition, given as a discrete number and unit of measurement |
| ☐ | ☒ | A statement on whether measurements were taken from distinct samples or whether the same sample was measured repeatedly |
| ☐ | ☒ | The statistical test(s) used AND whether they are one- or two-sided *Only common tests should be described solely by name; describe more complex techniques in the Methods section.* |
| ☐ | ☒ | A description of all covariates tested |
| ☐ | ☒ | A description of any assumptions or corrections, such as tests of normality and adjustment for multiple comparisons |
| ☐ | ☒ | A full description of the statistical parameters including central tendency (e.g. means) or other basic estimates (e.g. regression coefficient) AND variation (e.g. standard deviation) or associated estimates of uncertainty (e.g. confidence intervals) |
| ☐ | ☒ | For null hypothesis testing, the test statistic (e.g. *F*, *t*, *r*) with confidence intervals, effect sizes, degrees of freedom and *P* value noted *Give P values as exact values whenever suitable.* |
| ☒ | ☐ | For Bayesian analysis, information on the choice of priors and Markov chain Monte Carlo settings |
| ☒ | ☐ | For hierarchical and complex designs, identification of the appropriate level for tests and full reporting of outcomes |
| ☒ | ☐ | Estimates of effect sizes (e.g. Cohen's *d*, Pearson's *r*), indicating how they were calculated |

*Our web collection on statistics for biologists contains articles on many of the points above.*

## Software and code

Policy information about availability of computer code

| Data collection | CryoEM data was collected on a Titan Krios electron microscope, data collection was automated by the Thermofisher EPU (v1.9) software. |
|---|---|
| Data analysis | CryoEM data analysis utilised Relion2.1, MotionCor2 v1.2.0, Warp 1.0.4, cryoSPARC v2. Movies were motion corrected and dose weighted using MotionCor2 v1.2.0. Corrected images were imported into Warp 1.0.4 where particles were picked using a Warp box net. Particle stacks were imported into cryoSPARC v2 and were used in a homogenous refinement. The resulting structure from cryoSPARC v2 was then further refined in RELION 2.1.  SPIDER (version UNIX 26.04) was used for symmetry analysis. Models of cryoEM structures were visualised in UCSF ChimeraX 1.13.1rc.<br><br>Negative stain TEM: The periodicity of microfibril repeats was measured using ImageJ2.<br><br>STEM mass mapping: Mass per unit length measurements and axial mass distributions were measured from STEM ADF images using the Semper6 image analysis software (Synoptics).<br><br>STEM mass data and periodicity measurements were analysed in Graphpad Prism 9.1.2. |

For manuscripts utilizing custom algorithms or software that are central to the research but not yet described in published literature, software must be made available to editors and reviewers. We strongly encourage code deposition in a community repository (e.g. GitHub). See the Nature Portfolio guidelines for submitting code & software for further information.

# Data

Policy information about availability of data

All manuscripts must include a data availability statement. This statement should provide the following information, where applicable:
- Accession codes, unique identifiers, or web links for publicly available datasets
- A description of any restrictions on data availability
- For clinical datasets or third party data, please ensure that the statement adheres to our policy

The cryoEM data has been deposited to EMBD with accession codes EMD-13984 for the bead model and EMD-13986 for the arm region. The values plotted in figures 3 and 4 are provided as source data online. Any additional information required to reanalyse the data reported in this study is available from the corresponding author.

# Field-specific reporting

Please select the one below that is the best fit for your research. If you are not sure, read the appropriate sections before making your selection.

☒ Life sciences ☐ Behavioural & social sciences ☐ Ecological, evolutionary & environmental sciences

For a reference copy of the document with all sections, see nature.com/documents/nr-reporting-summary-flat.pdf

# Life sciences study design

All studies must disclose on these points even when the disclosure is negative.

| | |
|---|---|
| Sample size | No sample size calculations were performed. For cryoEM imaging of bovine microfibrils, at least six independent datasets were collected in the optimisation of the sample and data collection strategy. For the final cryoEM dataset, 1310 movies were collected and 27,737 microfibril periods were picked from the motion corrected images.<br><br>For negative stain or STEM mass mapping of microfibrils, 100 periods were measured. We determined this to be sufficient based on previous similar studies and low variability between samples doi: 10.1016/j.jmb.2010.04.008; doi: 10.1016/s1357-2725(97)00028-9.<br><br>Biochemical experiments were repeated at least three times, with each experiment containing thousands of molecules. Performing biochemical experiments in biological triplicate is a widely used replication standard. |
| Data exclusions | The cryoEM pipeline excludes "bad" particles in an automated software pipeline that is widely utilised in the field and is well-described. For all other data types, no data were excluded from analysis. |
| Replication | All attempts at replication were successful.<br>Biochemical experiments were repeated at least three times.<br>Negative staining imaging on wildtype and mutant microfibrils was repeated at least twice on each different tissue sample.<br>For cryoEM, more than six microfibril purifications were performed on six different ciliary zonule samples (biological replicates) for optimisation of grid preparation and data collection strategy. At least six cryoEM datasets were collected independently in the optimisation process and all data were consistent with the final data set. |
| Randomization | Randomization is not relevant for this study, as there were no groups allocated in any of the experiments. |
| Blinding | Blinding was not relevant for the cryoEM data as data was collected on only one sample type.<br>For STEM mass mapping and negative stain periodicity measurements, blinding was not performed as in each sample, images of all microfibrils were collected and from these images all microfibril periods were included. The same measurement procedures were used for all images and for the negative stain data particle picking and classification was automated to prevent bias. The readout from these data was a quantifiable parameter ie mass per repeat or microfibril periodicity, respectively, rather than a subjective observation. |

# Reporting for specific materials, systems and methods

We require information from authors about some types of materials, experimental systems and methods used in many studies. Here, indicate whether each material, system or method listed is relevant to your study. If you are not sure if a list item applies to your research, read the appropriate section before selecting a response.

## Materials & experimental systems

| n/a | Involved in the study |
|-----|----------------------|
| ☐ | ☒ Antibodies |
| ☒ | ☐ Eukaryotic cell lines |
| ☒ | ☐ Palaeontology and archaeology |
| ☐ | ☒ Animals and other organisms |
| ☒ | ☐ Human research participants |
| ☒ | ☐ Clinical data |
| ☒ | ☐ Dual use research of concern |

## Methods

| n/a | Involved in the study |
|-----|----------------------|
| ☒ | ☐ ChIP-seq |
| ☒ | ☐ Flow cytometry |
| ☒ | ☐ MRI-based neuroimaging |

## Antibodies

| | |
|---|---|
| Antibodies used | Commercial antibodies from Sigma Aldrich MAB2502 (Anti-Fibrillin-1 Antibody, clone 26; Lot number 0608038538) and MAB1919 (Anti-Fibrillin-1 Antibody, clone 11C1.3; Lot number 3256666 ). |
| Validation | MAB2502 recognizes human Fibrillin-1. Epitope mapping studies identify the binding site of this antibody to amino-terminal end of the molecule, between amino acid residues 45 and 450. The antibody is reactive with human, chicken, and bovine Fibrillin-1. MAB1919 recognizes human Fibrillin-1. The antibody is reactive with bovine, pig, and human Fibrillin-1. Publications DOI: 10.1242/jcs.029819, DOI: 10.1016/0945-053x(94)90028-0; DOI: 10.1074/jbc.M111.231571; DOI: 10.1006/jmbi.1996.0237 In the manuscript, both antibodies are used in western blotting against human fibrillin-1 constructs to further define their epitopes to map their binding location on fiibrilln microfibrils. |

## Animals and other organisms

Policy information about studies involving animals; ARRIVE guidelines recommended for reporting animal research

| | |
|---|---|
| Laboratory animals | Skin samples were collected from 6-week old male homozygous WMΔ mice, ΔHybrid1 mice or control (C57BL/6) mice |
| Wild animals | The study did not involve wild animals |
| Field-collected samples | The study did not involve samples collected from the field |
| Ethics oversight | Landesamt für Natur, Umwelt und Verbraucherschutz Nordrhein-Westfalen for breeding (permit No. 84-02.04.2014.A397) and euthanasia (permit No. 84-02.05.40.14.115). |

Note that full information on the approval of the study protocol must also be provided in the manuscript.

