## [Peer Review File · Nature Structural & Molecular Biology]

Peer Review Information

Manuscript Title: Fibrillin microfibril structure identifies long-range effects of inherited pathogenic mutations affecting a key regulatory latent TGF β -binding site

Corresponding author name(s): Clair Baldock

Reviewer Comments & Decisions:

Decision Letter, initial version:

Message: 25th May 2022

Dear Dr. Baldock,

Thank you again for submitting your manuscript "Fibrillin microfibril structure identifies long-range effects of inherited pathogenic mutations affecting a key regulatory TGF β -binding site". I sincerely apologize for the delay in responding, which resulted from the difficulty in obtaining suitable referee reports. Nevertheless, we now have comments (below) from the 3 reviewers who evaluated your paper. In light of those reports, we remain interested in your study and would like to see your response to the comments of the referees, in the form of a revised manuscript.

You will see that reviewers #2 and #3 raise some issues regarding the cryo-EM resolution and density interpretation. Please be sure to address/respond to all concerns of the referees in full in a point-by-point response and highlight all changes in the revised manuscript text file. If you have comments that are intended for editors only, please include those in a separate cover letter.

We expect to see your revised manuscript within 6 weeks. If you cannot send it within this time, please contact us to discuss an extension; we would still consider your revision, provided that no similar work has been accepted for publication at NSMB or published elsewhere.

As you already know, we put great emphasis on ensuring that the methods and statistics reported in our papers are correct and accurate. As such, if there are any changes that should be reported, please submit an updated version of the Reporting Summary along

with your revision.

Reporting Summary:

Please note that all key data shown in the main figures as cropped gels or blots should be presented in uncropped form, with molecular weight markers. These data can be aggregated into a single supplementary figure item. While these data can be displayed in a relatively informal style, they must refer back to the relevant figures. These data should be submitted with the final revision, as source data, prior to acceptance, but you may want to start putting it together at this point.

Data availability: this journal strongly supports public availability of data. All data used in accepted papers should be available via a public data repository, or alternatively, as Supplementary Information. If data can only be shared on request, please explain why in

your Data Availability Statement, and also in the correspondence with your editor. Please note that for some data types, deposition in a public repository is mandatory - more information on our data deposition policies and available repositories can be found below: <https://www.nature.com/nature-research/editorial-policies/reporting-standards#availability-of-data>

[Redacted]

Sincerely,
Sara

Sara Osman, Ph.D.
Associate Editor
Nature Structural & Molecular Biology

Referee expertise:

Referee #1: Connective tissue diseases

Referee #2: Structure-function studies of fibrils

Referee #3: Cryo-EM

Reviewers' Comments:

Reviewer #1:

Remarks to the Author:

The authors describe a cryoEM structure of bovine fibrillin-containing microfibrils. Microfibrils are present in many tissues and are at the root of several genetic disorders, such as Marfan syndrome. While several features are known from structural studies using negative staining TEM or SAXS, some additional features become evident at higher resolution. CryoEM of microfibrils is certainly a very difficult task due to structural variabilities and flexibility of these structures. High resolution was achieved for the bead region. The authors used this structure to position an N-terminal fibrillin-1 region into the bead region based on monoclonal antibody mapping from this and other groups, as well as on protease resistant regions in the N-terminus. The authors also include analyses of microfibrils isolated from genetic mouse models with deletions in the N-terminal region, which helped to define the respective region relative to the bead. Finally, they study LTBP-1 binding and the structural consequences and refined the interbead region from cryoEM data. Overall, the work is very interesting and important for the field. However, a series of aspects appear not developed sufficiently and thus several conclusions are not justified.

Major comments:

1. The authors describe the importance of fibrillin-containing microfibrils in relation to elastic fiber formation, disease involvement in this process, and microfibril-binding proteins. But then they isolate and work with microfibrils from ciliary zonules which are basically elastin-free microfibrils and very few binding proteins have been identified for those microfibrils. Ciliary zonules likely have an entirely mechanical function in the accommodation of the lens, whereas other microfibrils in blood vessels and bone likely have different functions and are presumably structurally somewhat different. It is clear that bovine ciliary zonules have been used, because it is feasible to purify them from this tissue. But the aspects above have not been made clear in the manuscript.
2. The authors claim in Fig. 1E for the bead region that individual fibrillin molecules can be seen. It is not clear in this figure how the authors distinguish between fibrillin and other potential proteins present in the bead. We already know from previous studies that there is a tight packing in the bead region.
3. The arm structure in Fig. 1A-C has been previously published by this group, albeit with a lower resolution (Godwin et al. JMB 430, 2018). The current analysis confirms the bead, arm, interbead regions and includes a new shoulder region. However, this shoulder region could be visualized in the previous paper, albeit it was not named like that. What is also better visible here are the details of the arms. But some of these aspects appear like a confirmation of previous data.
4. Fig 2A,B: No markers are included on the Western blots and no predicted molecular

masses are indicated for the recombinant fibrillin-1 fragments. This does not allow to validate whether or not the bands correlate with the expected masses. Also, no indication was provided how often these experiments were repeated. Some of the bands are very faint, thus it would be desirable to include another method for confirmation (e.g. ELISA).

5. The authors comment on page 7 that the binding sites of two monoclonal antibodies (mAb 1919, 2502) have been mapped previously to the arm region and bead region, respectively, citing refs 13 and 36. mAb 2502 used in ref 36 seems to be the same than mAb 26 used in ref 13. Ref 36 states that this antibody does recognize human but not bovine microfibrils. On page 7, however, the authors infer that this antibody binds to the bovine microfibrils. If the epitope is not present in bovine microfibrils, the structure is likely different compared to human microfibrils. The second antibody mAb 1919 has also a double name between papers (11C1.3). In the previous paper, the description was that this antibody binds "at the interbead striation where the arms terminate". However, this does not seem to match the position how it is situated in Fig. 2D. Overall for this point, there is quite a bit of confusion and inconsistency in terms of these antibodies and their interpretations and conclusions in the present paper.

6. Inspecting refs 13 and 20 for mapped antibody locations, it seems that mAb 26 (epitope in TB1) and mAb 201 (epitope in TB2 region) were previously mapped to microfibrils on the same side of the bead. The positioning of TB1 and TB2 relative to the bead as shown in Fig. 2D is not consistent with those results. The two antibodies in this model would map at the opposite sides of the bead, as well as too close to the beads, compared to previously published data.

7. The authors refer to a proteomic study for Fig. 2 showing the lack of peptides from domains EGF4-TB2 after elastase digestion (ref 38). The lack of peptides in this region could simply be a consequence of the lack of elastase recognition sites. To arrive at this conclusion with confidence, it would be necessary to perform such experiments with several proteases and more importantly with low specificity proteases such as trypsin or similar proteases to make sure that there are definitely cleavage sites in this region. Overall, the positioning of the fibrillin-1 stretch shown in Fig. 2D is not fully justified.

8. The authors used negative staining in Fig. 4 to analyze microfibrils from skin of mice with deletions in the N-terminal region. I understand that it is more difficult to obtain sufficient amounts of microfibrils from mouse skin and thus the resolution is much lower than that of cryoEM. The authors point out very minor differences close to the bead in these images. I am not convinced whether these differences are true differences or whether these are variations due to the low sample numbers and the lower resolution imaging technique. The authors indeed state in the discussion that this shoulder region was difficult to resolve with single particle averaging approaches suggesting that this region has conformational heterogeneity. It is not clear why SEM was used for the mean periodicity in 4B. This should be standard deviation. The figure 4 legend title indicates a disrupted bead region, but from the images and the text the disruption is in the shoulder region.

9. LTBP-1 binding in Fig. 5: The authors conclude from these studies long range structural changes in the WMS deletion construct. But the mapped LTBP-1 binding site is actually quite close to this deletion, just a few domains upstream. This is in my opinion not a long range structural change. From the existing work of fibrillin-1 domain structures, it is quite well established that domains in close vicinity can affect each other structurally. The mass

per repeat changes after LTBP-1 is bound. This would require a repetitive and regular binding. How do the authors reconcile these data with previous publications showing that LTBP-1 is not regularly present on microfibrils (PMID 12429738 – not cited in the manuscript). Is LTBP-1 present in ciliary zonules?

10. On page 13, the authors argue that MAGP1 is co-purified with the microfibrils. MAGP1 localizes on or close to the beads. Can the authors determine in the cryoEM structure where MAGP1 is located?

11. Figure 6D(ii): The authors used those antibody mappings that fit with their model. But importantly, mAb 201 is omitted which is expected to be on the same side relative to the bead than 2502. Again, this would not be consistent with the data, details are described above.

12. The title of the manuscript does not seem to be fully appropriate for what is shown in this paper. Also, there is no direct TGF-beta binding site in fibrillin-1.

Minor comments:

13. Page 3, 2nd paragraph: should read: .. with fibrillin-1 being the predominant form in the adult... [because it is not the predominant form in the embryo]

14. Page 4, 2nd paragraph: Binding of some integrins to fibrillin-1 has been much earlier identified by the Timpl and Mecham groups (PMIDs 8617364, 8617764) These should be cited as well.

15. The use of ImageJ should be cited as the developing team requests on their website.

Reviewer #2:

Remarks to the Author:

The 10 - 12 nm diameter fibrillin microfibrils play critical roles in connective tissue integrity and extracellular matrix growth factor regulation. The organisation of fibrillin within microfibrils has been an intractable problem since the discovery of fibrillins as the major components of microfibrils in 1986, and this information is needed to understand how microfibrils carry out their functions. This paper is very clearly written and presents novel information on the structure of microfibrils, at the highest resolution to date, and so has the potential to make a significant contribution to our understanding of how these structures are assembled and organised. The long-range changes in microfibril structure reported, caused by mutations such as the WMS deletion, are also an important observation that provide new insights into the mechanisms associated with growth factor regulation and disease pathogenesis. The data showing LTBP1 binding to microfibrils are important in showing that the extracted microfibrils have maintained some function as well as showing the relevance of changes in the structure near the bead to growth factor regulation. There are a few issues with the interpretation of the data, however, that I would recommend reviewing before publication.

The cryoEM structure presented was scanned for a region that remained connected at increasing threshold levels, and the region identified was found to be similar in structure to the SAXS data obtained for fibrillin-1 fragment PF2 (domains TB1-TB2). The authors

suggest that this region of the cryoEM structure spans domains TB1-cbEGF6 based on the SAXS data and the identification of a protease-resistant region between domains EGF4 and TB2. Considering that structures within the assembled microfibril may be different to that seen in solution by SAXS, and that it has previously been suggested (Kuo et al., 2007 J. Biol. Chem. 282:4007; Hubmacher et al., 2008 PNAS 105: 6548) that the C-terminal domains form the core of the microfibril, how confident are the authors that the blue structure in figure 3A isn't another part of fibrillin-1, for example domains TB7-cbEGF41? This combination of a TB domain followed by cbEGF repeats may give a similar structure to that expected for TB1-cbEGF6, (depending on the conformation of the proline-rich region). Also, is it possible that the protease-resistant sites in domains EF4-TB2 are masked through some other interactions instead of being buried in the bead, and that domains TB1-cbEGF6 might instead be found in the grey areas shown in figure 3E? This would still be consistent with the antibody mapping data using mAb2502 and mAb1919, which give clear constraints on the positions of domains TB1-PRR and cbEGF7-hyb2.

The data in figure 4 is interesting because it shows that fibrillin-1 mutations can lead to long-range structural effects in microfibrils that may affect growth factor binding, which follows directly from the positioning of domains TB1-cbEGF6 to the bead region. This may reflect my lack of experience with the technique, but would it be possible to repeat the cryoEM work with the WMS mutant microfibrils to get a more detailed look at the structure? This isn't essential for this paper but would be an interesting comparison of the wild type and mutant forms at high resolution, and may help confirm whether the structure identified in the bead is the TB1-cbEGF6 region since the WMS mutant lacks domains TB1-EGF4.

In the discussion, the authors mention that the model produced for the bead structure is consistent with the presence of a transglutaminase cross-link between domains cbEGF5 and TB7 as well as an interaction observed between the N-terminal region and domains TB5-TB6. I'm not sure that it's clear that the transglutaminase cross-link described by Qian et al. occurs within individual microfibrils, or if it occurs as part of the stabilisation of microfibril bundles as tissues mature, so its utility as a constraint for fibrillin organisation in microfibrils may be limited. There should also be some mention in the discussion on how the described model fits with other observations of interactions between the N and C termini of fibrillin-1, including the electron microscopy data from Lin et al. 2002 (J. Biol. Chem. 277:50795) showing head-to-tail alignment of monomers, and the protein interaction data in Marson et al. 2005., (J. Biol. Chem. 280:5013) and Yadin et al. 2013 (Structure 21:1743). Marson et al. show two distinct N-N interactions (one involving possible folding at the proline-rich region), and both Marson et al. and Yadin et al. describe interactions involving the N terminus and specific C-terminal cbEGF domains. There should also be some discussion on the work of Hubmacher et al., 2008 PNAS 105: 6548 in terms of how this relates to the positioning of the C-terminal domains in the model and their role in the early stages of microfibril assembly.

Some minor points:

In the Western blot in Figure 2 B(ii), the PF2 and PF5 lanes are repeated. Also, the PF5 samples appear to behave differently with a band appearing in the non-reducing lane of one sample (left-hand side) but not the other (right-hand side). This should be corrected.

Would it be possible to colour panel D in figure 3 to reflect its relationship to the structures in panel B (i.e., what is going where)?

The position of the mAb69 binding site in figure 6 suggests that this antibody recognises the bead region of the microfibril, but the data in Reinhardt et al. 1996 (J. Mol. Biol. 258: 104-116) shows that this antibody binds to the side of the beads. The mAb69 epitope is also more narrowly defined as being between domains TB6 and TB7 in Kuo et al 2007 J. Biol. Chem. 282:4007.

Reviewer #3:

Remarks to the Author:

This paper describes the structure of fibrillin microfibrils derived from single particle analysis of cryo-EM data combined with EM data from antibody labelling and mutated systems. Fibrin microfibrils represent an important system of considerable biological and medical relevance and establishing their detailed structure is of widespread interest. The paper which is well written and clearly presented describes a number of novel structural insights. Nevertheless, the 3D map of the fibrillin bead region, which represents the major finding of the current work is rather limited in detail and this introduces significant ambiguity into the interpretation.

Detailed points.

1) The authors apply C2 symmetry in their 3D analysis of the fibrillin bead region. This should be justified. It is important to avoid applying a pseudosymmetry which would result in loss of information. It might be useful to conduct an independent analysis with no applied symmetry and comparing this with the C2 map as an aid to clarifying the situation. The authors also identify a C8 pseudosymmetry of fibrillin monomers, but avoid applying C8.

2) The authors interpret extended densities in their bead region map as the N-terminal regions of fibrillin monomers. The map is itself quite complex being described by the authors as an interwoven structure. In this situation it can be quite tricky to correctly identify the correct connectivity of an individual extended molecule such as fibrillin. The authors assignment appears to have been based on identifying a feature which retains its connectivity at high threshold. I'm not sure that this is necessarily a reliable criterion for assigning protein density. I think the assignment of well resolved arm-like filamentous densities away from the body of the bead to fibrillin is reasonable. However, within the bead itself it there appears to be quite a complex network of densities and it would help to have some additional support for the current interpretation or perhaps to critically evaluate alternative arrangements.

3) Having assigned the chosen density regions to fibrillin molecules the authors compare these regions to a SAXS based structural model. This is illustrated for one of the fibrillin molecules in figure 3c. Here the overall dimensions and curvature of the segmented region of the map matches the model quite well. However, the correspondence between the map and the assigned component domains of fibrillin is very limited. I think this needs to be discussed within the context of the resolution of the analysis. The authors estimate an overall resolution of 9.7 Angstroms to their map of the microfibril bead. At this resolution I would expect the densities for the individual EGF and TB domains to closely match their model structures which is clearly not the case in figure 3c. Perhaps an analysis of local resolution might clarify this apparent contradiction.

4) On page 9 the authors suggest that flexibility in the arm domains may have led to loss of density in their analysis. In the case of conformational variability of this nature it may be helpful to explore various types of classification (local and global) in order to resolve individual conformers. Was this attempted?

Author Rebuttal to Initial comments**Response to reviewers' comments**

The authors thank the reviewers for their helpful comments and suggestions. We have addressed each point as follows.

Reviewer #1:

Major comments:

1. microfibrils from ciliary zonules which are basically elastin-free microfibrils and very few binding proteins have been identified for those microfibrils. Ciliary zonules likely have an entirely mechanical function in the accommodation of the lens, whereas other microfibrils in blood vessels and bone likely have different functions and are presumably structurally somewhat different. It is clear that bovine ciliary zonules have been used, because it is feasible to purify them but the aspects above have not been made clear in the manuscript.

The reviewer is correct in that the microfibrils used for cryoEM are from bovine ciliary zonules. This is because the ciliary zonule is a very pure source of fibrillin microfibrils and so purification does not require the use of proteases which can degrade the microfibrils. We have made this point clearer at the start of the results section by adding a sentence "Microfibrils were isolated from bovine ciliary zonule as it is a very pure source of fibrillin microfibrils and so purification does not require the use of proteases which can degrade the microfibrils." There are some microfibril binding proteins in CZ microfibril preparations including MAGP1, EMILIN, LTBP2, ADAMTSL4 and fibulin-2 (De Maria et al., 2018 and confirmed with our own analysis) but the reviewer is correct that elastin is not found in the ciliary zonules. Our ultrastructural analysis of microfibrils from different tissues has found that the microfibril backbone is consistent between tissues with only minor differences. This backbone should serve as a conserved deposition scaffold for tropoelastin and adaptor molecules such as LTBP2 or fibulins where the amounts of microfibril-binding proteins may differ in a tissue-specific manner. However, this study is aimed at getting insight into the fundamental microfibril structure and therefore, ciliary zonule derived microfibrils are well suited for this structural investigation as they can be purified without protease digestion; they only contain fibrillin-1 and have a limited number of microfibril associated-proteins. We have added this point to the first paragraph of the discussion to make this point clearer.

2. The authors claim in Fig. 1E for the bead region that individual fibrillin molecules can be seen. It is not clear in this figure how the authors distinguish between fibrillin and other potential proteins present in the bead.

The 3D reconstruction of the bead region is constructed from more than 7,000 microfibril "particles" so the density observed in the reconstruction is present in all 7,000 microfibrils that contribute to the reconstruction. It is unlikely that a binding partner would be present in every microfibril and therefore wouldn't contribute to the reconstruction. For this reason, this is why we think this core microfibril structure is composed of fibrillin, and as microfibril associated proteins are likely present in a low stoichiometry (suggested from MS peptide count compared to fibrillin

peptides), they wouldn't be detected in the overall reconstruction. We've now added a sentence on page 6 of the manuscript to explain this point.

3. The arm structure in Fig. 1A-C has been previously published by this group, albeit with a lower resolution (Godwin et al. JMB 430, 2018). The current analysis confirms the bead, arm, interbead regions and includes a new shoulder region. However, this shoulder region could be visualized in the previous paper, albeit it was not named like that. What is also better visible here are the details of the arms. But some of these aspects appear like a confirmation of previous data.

Our previous paper combined serial blockface SEM, electron tomography and negative stain TEM to understand the hierarchical organisation of fibrillin microfibrils in ciliary zonules. Negative staining is associated with artefacts including dehydration and flattening of the sample. Here we report the first frozen hydrated structure of a fibrillin microfibril (or indeed any extracellular matrix fibrillar polymer) which for the first time allows visualisation of internal structure of the microfibril at sub-nanometre resolution. In our previous paper, we showed the interbead region as the point between two beads (following the arms), and the shoulder region is adjacent to that (a shoulder to the bead), but sometimes the term interbead is used to mean the region between two beads. We've checked our terminology to make sure it is consistent with the labelling shown in figure 6D where the bead, arms, interbead and shoulder are labelled.

4. Fig 2A,B: No markers are included on the Western blots and no predicted molecular masses are indicated for the recombinant fibrillin-1 fragments. This does not allow to validate whether or not the bands correlate with the expected masses. Also, no indication was provided how often these experiments were repeated. Some of the bands are very faint, thus it would be desirable to include another method for confirmation (e.g. ELISA).

We apologise for the omission of markers and clear labelling of the western blots. We have repeated this experiment and included new blots that are clearly labelled and we have indicated the expected size of the protein constructs.

5. mAb 2502 used in ref 36 seems to be the same than mAb 26 used in ref 13. Ref 36 states that this antibody does recognize human but not bovine microfibrils. On page 7, however, the authors infer that this antibody binds to the bovine microfibrils.

It was not our intention to infer that this antibody bound to bovine microfibrils, indeed we did not test this in this study. The western blotting, described on page 7 was with human fibrillin-1 fragments and this has now been clarified in the text.

If the epitope is not present in bovine microfibrils, the structure is likely different compared to human microfibrils.

Human and bovine fibrillin-1 have very similar amino acid sequences with 98% sequence identity. The region we have indicated that mAb 2502 recognises is 92% identical between

human and bovine, but presumably small differences in sequence in the epitope are sufficient that the antibody does not recognise bovine microfibrils.

92.5% identity in 120 residues overlap; Score: 642.0; Gap frequency: 0.0%

```
UserSeq1      1  DVRPGYCYTALTNGRCSNQLPQSITKMQCCCDAGRCWSPGVTVAPEMCP IRATEDFNKLC
UserSeq2      1  DVRPGYCYTALANGRCSNQLPQSITKMQCCCDVGRWCWSPGVTVAPEMCP IRATEDFNKLC
*****
```

```
UserSeq1     61  SVPMVIPGRPEYPPPPLGPIPPVLPVPPGFPFPGPQIPVPRPPVEYLYPSREPPRVLPVNV
UserSeq2     61  SVPMVIPERPGYPPPPLGVPVPPVQVPPGFPFPGPQIMIPRPPVEYPYPSREPPRVLPVNV
*****
```

The second antibody mAb 1919 has also a double name between papers (11C1.3). In the previous paper, the description was that this antibody binds “at the interbead striation where the arms terminate”. However, this does not seem to match the position in Fig. 2D.

What we referred to as an interbead striation from earlier negative stain images is the double-band where the arms meet the interbead that is highlighted in figure 6 panel C. Docking into the cryoEM structure suggests that domains Hybrid2 and TB2 occupy this striation, which would be consistent with the location of the mab1919/11C1.3 epitope now mapped to within domains EGF-Hybrid2. Also, to note that when imaging antibody positions, an IgG molecule is ~16 nm long so there will be ~8nm between its centre of mass and the binding epitope. 8nm is approximately the length of 3-4 fibrillin-1 domains so the mapping accuracy of binding location will only be within a few domains. We modified the cartoon in figure 2D as this simplified representation did show the antibody and domain location offset to the left.

6. Inspecting refs 13 and 20 for mapped antibody locations, it seems that mAb 26 (epitope in TB1) and mAb 201 (epitope in TB2 region) were previously mapped to microfibrils on the same side of the bead. The positioning of TB1 and TB2 relative to the bead as shown in Fig. 2D is not consistent with those results. The two antibodies in this model would map at the opposite sides of the bead, as well as too close to the beads, compared to published data.

In our previous paper, Baldock et al JCB 2001 (ref 36), we mapped mAb2502 (mAb26) binding to the opposite side of the bead to mAb 11C1.3 (1919) which is why we referred to the locations of these antibodies. Mab 11C1.3/1919 has a similar epitope to mab 201, binding to domains immediately downstream of TB2. Our observation was different to the findings of Reinhardt et al., JMB 1996 (ref 13) where the double-labelling of mAb26 and mAb201 indicated that mAb26 and mAb201 are on the same side of the bead. In trying to reconcile this difference, we considered the samples used for these studies, where purified adult human ciliary zonule microfibrils were used in ref 36 and microfibrils in human neonate foreskin tissue were immuno-labelled enbloc in ref 13. There are data supporting conformational changes and microfibril maturation over time, where some antibodies only recognise a mature fibrillin microfibril matrix in culture (ref 36). We considered whether the N-terminus could be flexible, particularly in developing or immature microfibrils, which could explain these differences in the location of the TB1 domain. Given the differences in the data on the position of mAb 2502, and as suggested by reviewer 2, we have

provided an alternative path through the bead region, where the C-terminal region could contribute to the central bead core, with the N-terminus occupying the outer layer of the bead. In this arrangement, perhaps conformational variability would enable the TB1 domain to occupy a position closer to TB2 which would reconcile both antibody mapping observations. We have now modified the legend to figure 2D to indicate these are the relative locations of mAbs 2502 and 1919 in adult CZ microfibrils, we've modified figure 3 to add an additional panel showing an alternate path through the bead and added to the text details of this arrangement making reference to differences in mAb2502/26 location (new paragraph on pages 9/10). We have also removed panel D from figure 6, which showed possible domain positioning in microfibril packing models. Instead, in a new figure 8 we have removed reference to packing models and antibody mapping, and instead provided simplified schematics of microfibril assembly and overlaid domain assignments from the domain deletions, docking and integrin $\alpha V\beta 3$ binding (new figure 7).

7. The authors refer to a proteomic study for Fig. 2 showing the lack of peptides from domains EGF4-TB2 after elastase digestion (ref 38). The lack of peptides in this region could simply be a consequence of the lack of elastase recognition sites. To arrive at this conclusion with confidence, it would be necessary to perform such experiments with several proteases and more importantly with low specificity proteases such as trypsin or similar proteases to make sure that there are definitely cleavage sites in this region. Overall, the positioning of the fibrillin-1 stretch shown in Fig. 2D is not fully justified.

Eckersley et al, JBC (ref 38) made a detailed analysis of the comparative peptide hits from elastase and Smart digestion (which utilises trypsin) to determine which approach yields more peptides from fibrillin microfibrils from different tissues. This paper reported that elastase digestion was more successful than trypsin in yielding more peptide hits for fibrillin. Indeed, in the region from EGF4-TB2 there are 29 predicted cleavage sites for elastase and 26 for trypsin so the lack of peptides from this region is not due to lack of protease recognition. Furthermore, studies by Cain et al 2006 which used trypsin rarely detected peptides from this region which suggests in non-denaturing conditions, it is inaccessible to proteolytic enzymes and buried within the microfibril structure. When microfibrils are extracted in denaturing conditions, peptides from this region are present (De Maria et al, 2017) which suggests that this region is present but buried and inaccessible in native conditions. As mentioned in the response above we have also offered an alternative scenario in figure 3C(ii) where the C-terminal region may occupy this central core and N-terminal region be on the outside of the bead.

8. The authors used negative staining in Fig. 4 to analyze microfibrils from skin of mice with deletions in the N-terminal region. The authors point out very minor differences close to the bead in these images. I am not convinced whether these differences are true differences or whether these are variations due to the low sample numbers and the lower resolution imaging technique. The authors indeed state in the discussion that this shoulder region was difficult to resolve with single particle averaging approaches suggesting that this region has conformational heterogeneity.

Although negative stain is lower resolution, it is much higher contrast than cryoEM and in an averaged image of >450 repeats, it shows the main microfibril features are conserved between all samples. The negative staining is also sensitive enough to show differences in periodicity commensurate with the absence of 1 or 3 domains, respectively. In the averaged images, the main difference was in the shoulder region. This region has conformational flexibility that prevents high-resolution reconstruction by cryoEM due to the challenges in aligning this region with Angstrom precision in low contrast images. However, the increased contrast in negative stain images and nanometre-resolution means that class averages of the shoulder region from control microfibrils shows a discrete banding which is absent in both mutants so we are confident in the differences observed in figure 4 in the mutant microfibrils.

It is not clear why SEM was used for the mean periodicity in 4B. This should be standard deviation.

We have changed the chart in figure 4B to show the error bars as standard deviation.

The figure 4 legend title indicates a disrupted bead region, but from the images and the text the disruption is in the shoulder region.

This has now been corrected.

9. LTBP-1 binding in Fig. 5: The authors conclude from these studies long range structural changes in the WMS deletion construct. But the mapped LTBP-1 binding site is actually quite close to this deletion, just a few domains upstream. This is in my opinion not a long range structural change. From the existing work of fibrillin-1 domain structures, it is quite well established that domains in close vicinity can affect each other structurally. The mass per repeat changes after LTBP-1 is bound. This would require a repetitive and regular binding. How do the authors reconcile these data with previous publications showing that LTBP-1 is not regularly present on microfibrils (PMID 12429738 – not cited in the manuscript). Is LTBP-1 present in ciliary zonules?

The LTBP1 binding site (EGF2-3-hybrid1) is 2-4 domains upstream of the three domains deleted in WMS (TB1-PRR-EGF). This correlates to a length-scale of ~50-100 Angstrom, which in terms of protein structure is a long-range perturbation ie not within the one domain but effecting a distal interaction. We have reworded the text on page 14 to make this clearer "The LTBP-1 binding epitope is 2-4 domains upstream of the domains deleted in WMS [27], this further supports longer-range structural rearrangements perturbing a binding site at least 50 Å away when these domains are deleted."

For the experiment showing LTBP1 binding to microfibrils, we added exogenous LTBP1 to microfibrils to show where the LTBP1 binding site is on the microfibril, and the stoichiometry of binding when added at a molar excess which results in regular, repetitive binding to microfibrils. We agree that LTBP1 is not an integral component of a microfibril (Isogai et al., 2003 – citation now included in the manuscript and stated on page 14 that LTBP-1 is not an integral microfibril component), rather that it can bind to microfibrils and we can show where the binding site is located. This is in agreement with Isogai et al and Ono et al (ref 25) who showed that recombinant fibrillin molecules bind to LTBP1, but to the best of our knowledge, this is the first time it has been

shown that LTBP1 also binds to assembled microfibrils. LTBP1 is not detected in MS analysis of our microfibril purifications, consistent with published data that indicate it is only present as a very minor component of ciliary zonules (0.025%) (De Maria et al IOVS 2017), indicating that the LTBP1-binding sites would be unoccupied. We have now added a sentence the results “LTBP1 is not present (or only at very low abundance) in the ciliary zonules and so does not co-purify with these microfibrils”. We also mention in the discussion, that LTBP2 is present in our microfibril purifications, so some of the LTBP binding sites may be occupied by LTBP2, which could be why the stoichiometry is not higher.

10. On page 13, the authors argue that MAGP1 is co-purified with the microfibrils. MAGP1 localizes on or close to the beads. Can the authors determine in the cryoEM structure where MAGP1 is located?

Published MS datasets show the presence of MAGP1 (MFAP2) in a range of tissues including ciliary zonules, and we detect peptides from MAGP1 in our samples so it does co-purify with microfibrils, consistent with studies that have shown it is a major antigen associated with microfibrils (Gibson et al., 1986)(references now added to page 13). However, MAGP1 is only 19 kDa, in comparison with ~300 kDa for a fibrillin molecule and 2.5 MDa for a microfibril period. We assume that MAGP1 is present in the microfibril but cannot be visualised in the reconstruction due to its size. This is consistent with our previous unpublished observations where single particle class averages of microfibrils from MAGP1^{-/-} mice are indistinguishable from WT microfibrils. The AlphaFold prediction of the structure of MAGP1 suggests that over half of the molecule is unstructured and as such this inherent flexibility would not contribute to the 3D reconstruction. We've now added a sentence of page 6 to explain that we don't observe MAGP1 in the structure.

11. Figure 6D(ii): The authors used those antibody mappings that fit with their model. But importantly, mAb 201 is omitted which is expected to be on the same side relative to the bead than 2502. Again, this would not be consistent with the data, details are described above.

Also see response to point 6 above. We have removed panel D from figure 6, which showed possible domain positioning in microfibril models. In new figure 8 we have instead overlaid domain assignments from the domain deletions, docking and integrin $\alpha V\beta 3$ binding and removed reference to packing models and antibody mapping.

12. The title of the manuscript does not seem to be fully appropriate for what is shown in this paper. Also, there is no direct TGF-beta binding site in fibrillin-1.

We have modified the title to specify latent TGF β ie: Fibrillin microfibril structure identifies long-range effects of inherited pathogenic mutations affecting a key regulatory latent TGF β -binding site.

Minor comments:

13. Page 3, 2nd paragraph: should read: .. with fibrillin-1 being the predominant form in the adult... [because it is not the predominant form in the embryo]

This has been added

14. Page 4, 2nd paragraph: Binding of some integrins to fibrillin-1 has been much earlier identified by the Timpl and Mecham groups (PMIDs 8617364, 8617764) These should be cited as well.

These citations have been added

15. The use of ImageJ should be cited as the developing team requests on their website.

This citation has been added

Reviewer #2:

Considering that structures within the assembled microfibril may be different to that seen in solution by SAXS, and that it has previously been suggested (Kuo et al., 2007 JBC 282:4007; Hubmacher et al., 2008 PNAS 105: 6548) that the C-terminal domains form the core of the microfibril, how confident are the authors that the blue structure in figure 3A isn't another part of fibrillin-1, for example domains TB7-cbEGF41? Also, is it possible that the protease-resistant sites in domains EF4-TB2 are masked through some other interactions instead of being buried in the bead, and that domains TB1-cbEGF6 might instead be found in the grey areas shown in figure 3E? This would still be consistent with the antibody mapping data using mAb2502 and mAb1919, which give clear constraints on the positions of domains TB1-PRR and cbEGF7-hyb2.

At this resolution, it is challenging to unambiguously identify domain positioning on docking alone, which is why we utilised data from mouse models with domain deletions and antibody mapping data in our modelling. As we explain in our response to reviewer 3 below, the density protruding from the bead is connected and contiguous with the arm region, so we are confident with our assignment of domains cbEGF5 and 6 to this density. Also, on the underside of the bead (shown in grey in panel 3E) is an interwoven base which could form a platform for interactions to form, so there are other structures in the bead playing an important role that could be the C-terminal region. However, we acknowledge that it is possible that the domains preceding cbEGF5 and 6 could take a different path through or around the bead. We have therefore presented an alternative scenario where the core of the bead contains N- and C-terminal interacting regions in Figure 3C(ii). We have added an additional paragraph in the results text "Due to the dense molecular packing in the bead region and lower local resolution in the bead core region (Supplementary Figure 5), it is possible that the molecules in the core of the bead take a different path through the centre of the bead. Antibody binding data have previously shown that both the N- and C-terminal regions of fibrillin are also located near to the bead [13, 38] and the C-terminal half of fibrillin-1 has been shown to self-assemble into "bead-like" multimers which has been suggested to initiate microfibril assembly [16]. Therefore, we modelled an alternate path through the bead core, where the central core may be composed of N- and C-terminal interacting regions (Figure 3Cii) consistent with antibody mapping data [13, 38]." We have also added a new schematic representation in figure 8, where we state the C-terminal region plays an important role in assembly (citing Hubmacher et al., 2008) with a cartoon representation of the N- and C-terminal regions intertwined in the bead.

The data in figure 4 is interesting, but would it be possible to repeat the cryoEM work with the WMS mutant microfibrils to get a more detailed look at the structure?

CryoEM imaging is performed on microfibrils from bovine ciliary zonules as it is a rich source of relatively pure microfibrils, even so the yield and number of microfibrils we can image is low compared to the amount of particles used in a typical cryoEM dataset. From mice, due to the different anatomy of the eye and very small ciliary zonule, we are unable to extract microfibrils from this tissue. Therefore, we extracted microfibrils from elastin/collagen VI containing tissue, in this case skin, which requires digestion by collagenase enzymes and even then unfortunately does not yield sufficient microfibrils for cryoEM data collection. Moreover, microfibrils from skin co-purify with collagen VI microfibrils which are very difficult to differentiate between fibrillin microfibrils in cryoEM images. Although this is an excellent suggestion, it is not technically possible to perform this experiment using cryoEM for these reasons.

In the discussion... I'm not sure that it's clear that the transglutaminase cross-link described by Qian et al. occurs within individual microfibrils, or if it occurs as part of the stabilisation of microfibril bundles as tissues mature, so its utility as a constraint for fibrillin organisation in microfibrils may be limited. There should also be some mention in the discussion on how the described model fits with other observations of interactions between the N and C termini of fibrillin-1, including the electron microscopy data from Lin et al. 2002 (JBC 277:50795) showing head-to-tail alignment of monomers, and the protein interaction data in Marson et al. 2005., (JBC 280:5013) and Yadin et al. 2013 (Structure 21:1743). Marson et al. show two distinct N-N interactions (one involving possible folding at the proline-rich region), and both Marson et al. and Yadin et al. describe interactions involving the N terminus and specific C-terminal cbEGF domains. There should also be some discussion on the work of Hubmacher et al., 2008 PNAS 105: 6548 in terms of how this relates to the positioning of the C-terminal domains in the model and their role in the early stages of microfibril assembly.

We have amended the discussion and the model figure (was figure 6D, now new figure 8), to suggest how the interactions mapped using recombinant protein fragments could occur in microfibril assembly and/or within the structure of the mature microfibril. We've included the suggested references, where initial steps may be head-to-tail assembly mediated by the very terminal domains as described by Yadin et al. We think the N- and C-terminal interactions within the bead are more extensive than just the terminal domains, so secondary events may then give rise to more extensive N- and C-terminal interactions, consistent with data from Marson et al 2005 and Chaudhry et al. 2007. Certainly, lateral assembly and multimerisation to form the microfibril are important events and the C-terminal region seems to play an important role in this so we have suggested an alternative path through the bead in Figure 3Cii, as mentioned above, where the C-terminal domains could form the core of the bead which would be consistent with the behaviour of the recombinant C-terminal region as visualised by Hubmacher et al 2008.

Some minor points:

In the Western blot in Figure 2 B(ii), the PF2 and PF5 lanes are repeated. Also, the PF5 samples appear to behave differently with a band appearing in the non-reducing lane of one sample (left-hand side) but not the other (right-hand side). This should be corrected.

We have repeated the western blotting, also in response to review 1 point 7.

Would it be possible to colour panel D in figure 3 to reflect its relationship to the structures in panel B (i.e., what is going where)?

We have recoloured panels D and E in figure 3 following this suggestion.

The position of the mAb69 binding site in figure 6 suggests that this antibody recognises the bead region of the microfibril, but the data in Reinhardt et al. 1996 (JMB 258: 104-116) shows that this antibody binds to the side of the beads. The mAb69 epitope is also more narrowly defined as being between domains TB6 and TB7 in Kuo et al 2007 JBC 282:4007.

We have removed panel D from figure 6 and instead included a new figure 8 which does not show antibody mapping.

Reviewer

#3:

1) The authors apply C2 symmetry in their 3D analysis of the fibrillin bead region. This should be justified. It is important to avoid applying a pseudosymmetry which would result in loss of information. It might be useful to conduct an independent analysis with no applied symmetry and comparing this with the C2 map as an aid to clarifying the situation. The authors also identify a C8 pseudosymmetry of fibrillin monomers, but avoid applying C8.

In our previous analysis of fibrillin microfibrils using negative staining and TEM, we showed that different regions of the microfibril had symmetry. We analysed the sub-structures within the microfibril for symmetry, and rotational cross-correlation plots of the sub-structures for the bead, arm and interbead regions revealed higher-order pseudo-symmetries within the microfibril. Consequently, these regions were further refined applying 2-fold symmetry along the microfibril axis (Godwin et al., JMB 2018). We have checked in cryoEM that the symmetry is still present, we generated orientation correlation plots of C1 and C2 maps by rotating each about the z/symmetry axis. The initial C1 map had a self-correlation of 0.81 at 180 degrees which supported 2-fold symmetry.

We used C2 symmetry for the final bead map. The rotational cross-correlation plot of the C2 map shows the imposed 2-fold symmetry as a self-correlation of 1.0 when rotated by 180 degrees, and weaker 8-fold symmetry relationships are also apparent. A more stringent orientation analysis comparing 2 independent half maps, rotated at different relative orientations about the symmetry axis, gave a correlation of 0.84 at 180 degrees (and at zero degrees).

The lower correlations and less well-defined peaks indicate that the 8-fold is a pseudo rather than a true symmetry; and processing the data in higher symmetries led to degradation of features in the map and reduced resolution. We have now included these data in a new supplementary figure 4 and added reference to this in the results and methods sections.

2) The authors interpret extended densities in their bead region map as the N-terminal regions of fibrillin monomers. The map is itself quite complex being described by the authors as an interwoven structure. In this situation it can be quite tricky to correctly identify the correct connectivity of an individual extended molecule such as fibrillin. The authors assignment appears to have been based on identifying a feature which retains its connectivity at high threshold. I'm not sure that this is necessarily a reliable criterion for assigning protein density. It would help to have some additional support for the current interpretation or perhaps to critically evaluate alternative arrangements.

We agree with the reviewer that it is tricky to thread the molecule through the densely packed structure of the bead. However, we can see the density in the lower part of the bead (where we have docked domains cbEGF5 and 6) connecting to and contiguous with the arm region where docking continuous with domain TB2 onwards (as seen in Fig 6C). Docking the upstream domains from cbEGF4 to TB1 into the bead core positions domain TB1-PRR on the other side of the bead consistent with both antibody labelling and the sites of perturbation for WMS (TB1-PRR-EGF) and hybrid1 domain deletions (Figure 4). This docking is also consistent with our published antibody binding data for the downstream domains and the binding site for integrin $\alpha V\beta 3$ in domain TB4 in the interbead of the microfibril (new data in Figure 7). Nevertheless, at this resolution there could be some ambiguity in assigning the domain path and reviewer 2 makes a good point regarding the relative location of the N- and C-terminal regions in the bead and the path they may take. Therefore, although our docking in Figure 3C(i) is our preferred model based on other experimental data, we have indicated in a new figure panel 3C(ii), that there could be an alternative path around the outer-layer of the bead. This other arrangement would still have domains cbEGF5 and 6 connecting to the arm region and not change the arrangement of the molecule within the microfibril and still correlate with the locations of functional epitopes ie latent TGF β , the WMS-disease causing deletion and integrin binding.

3) Having assigned the chosen density regions to fibrillin molecules the authors compare these regions to a SAXS based structural model. This is illustrated for one of the fibrillin molecules in figure 3c. Here the overall dimensions and curvature of the segmented region of the map matches the model quite well. However, the correspondence between the map and the assigned component domains of fibrillin is very limited. I think this needs to be discussed within the context of the resolution of the analysis. The authors estimate an overall resolution of 9.7 Angstroms to their map of the microfibril bead. At this resolution I would expect the densities for the individual EGF and TB domains to closely match their model structures which is clearly not the case in figure 3c. Perhaps an analysis of local resolution might clarify this apparent contradiction.

At this resolution, the EGF and TB domains appear quite similar, they are both small domains with limited secondary structure. Analysis of the local resolution with cryoSPARC suggests that the resolution of the inner core of the bead is lower than the outer regions, which could explain the lack of more detailed features in this region. We have included the analysis of local resolution in a new supplementary figure 5 and in the results page 9 we have added an additional paragraph which mentions the lower local resolution in the bead core and possibility of alternate pathway around the bead (see also response to reviewer 2).

4) On page 9 the authors suggest that flexibility in the arm domains may have led to loss of density in their analysis. In the case of conformational variability of this nature it may be helpful to explore various types of classification (local and global) in order to resolve individual conformers. Was this attempted?

Yes, we did try local classification of the bead and arm regions to see if alternative conformations were present. Local 3D classification was performed in Relion but did not result in any classes which were grossly different in structure. However, this does not necessarily exclude conformational variability, as the overall data set had a relatively small number of particles in the final refinement, which meant that each class only had a few constituent particles which led to the individual 3D classes being much lower resolution.

Decision Letter, first revision:

Message: Our ref: NSMB-A45775A

30th Sep 2022

Dear Dr. Baldock,

Thank you for submitting your revised manuscript "Fibrillin microfibril structure identifies long-range effects of inherited pathogenic mutations affecting a key regulatory latent TGF β -binding site" (NSMB-A45775A). It has now been seen by the original referees and their comments are below. The reviewers find that the paper has improved in revision, and therefore we'll be happy in principle to publish it in Nature Structural & Molecular Biology, pending minor revisions to satisfy the referees' final requests and to comply with our editorial and formatting guidelines.

We are now performing detailed checks on your paper and will send you a checklist detailing our editorial and formatting requirements in about two weeks. Please do not upload the final materials and make any revisions until you receive this additional information from us.

To facilitate our work at this stage, we would appreciate if you could send us the main text as a word file. Please make sure to copy the NSMB account (cc'ed above).

Sincerely,
Sara

Sara Osman, Ph.D.
Associate Editor
Nature Structural & Molecular Biology

Reviewer #1 (Remarks to the Author):

All the criticisms have been adequately addressed in the revision.

Reviewer #2 (Remarks to the Author):

In the revisions of this paper, the authors have thoroughly addressed the concerns I raised in my initial review. However, there are just a few minor points that I think should be looked at before publication.

In the abstract, on line 34, the authors mention that this work reveals the "detailed cryo-EM structure of native fibrillin microfibrils". While the work is novel, and a much-needed contribution to our understanding of the structure of microfibrils, the statement suggests a higher level of resolution than has been achieved. I would perhaps tone this down to something like "a detailed structural analysis of native microfibrils by cryo-EM".

On line 86 of the introduction, I would replace "syndecans" with "heparan sulphate proteoglycans" because, although this has been strongly suggested, no direct observation of an interaction between syndecans and fibrillins has been made (at least as far as I know).

In the results section, in the work to narrow down the epitope of the mAb2502 antibody, it's not clear why domain EGF4 has been left out of the antibody binding site. According to the data presented, all of the fibrillin-1 fragments that are recognised by mAb2502 contain EGF4 but it has been left out of the red box in Figure 2. Similarly, it's not clear from the information provided why domains cbEGF10 - TB3 have been left out of the epitope mapped for mAb1919. If there is data from other work supporting this then it might be worth referring to it more explicitly for extra clarity.

In the legend to figure 3, line 270, I think there may be a mix up. It says that an alternative representation, where the C-terminal region (red) could be the inner core, is on the left of panel C (ii), but in the figure it is shown on the right.

In figure 7, in the bottom panel showing where the integrin headpiece seems to be binding to microfibrils, I would suggest increasing the size of the arrows indicating the binding site. As it is, the red marks appear to be very close to the bright region of the EM image (which seems to be another bead), which makes it confusing when you then look at figure 8 to find that the integrin binding TB4 domain is clearly located much further into the interbead region.

In figure 8, there's mention of domains cbEGF43-45 (line 473), which should be corrected (changed to cbEGF41-43) as fibrillin only has 43 cbEGF domains. This is also seen at the top of the image in figure 8 where there's mention of an interaction between FUN-EGF1 and "cbEGF43-45".

Reviewer #3 (Remarks to the Author):

The authors have provide detailed and thorough responses to all of the issues raised by the reviewers and have modified their manuscript appropriately. In a number of instances this has led them to revise their conclusions to include alternative explanations for their data to those described in the original paper. This is scientifically correct: however, it does mean that the story they tell is less clear cut. Overall I am happy for the paper to published in its current form

Author Rebuttal, first revision:

We're delighted that you are happy in principle to publish our manuscript entitled "Fibrillin microfibril structure identifies long-range effects of inherited pathogenic mutations affecting a key regulatory latent TGF β -binding site" in Nature Structural & Molecular Biology, pending minor revisions. We've made the minor text changes suggested by reviewer 2 to our manuscript which are detailed below.

In the abstract, on line 34, the authors mention that this work reveals the "detailed cryo-EM structure of native fibrillin microfibrils".... I would perhaps tone this down to something like "a detailed structural analysis of native microfibrils by cryo-EM".

We have changed the wording in the abstract to that suggested by the reviewer

On line 86 of the introduction, I would replace "syndecans" with "heparan sulphate proteoglycans" because, although this has been strongly suggested, no direct observation of an interaction between syndecans and fibrillins has been made.

We have replaced "syndecans" with "heparan sulphate proteoglycans" as suggested

In the results section, in the work to narrow down the epitope of the mAb2502 antibody, it's not clear why domain EGF4 has been left out of the antibody binding site. ... Similarly, it's not clear from the information provided why domains cbEGF10 - TB3 have been left out of the epitope mapped for mAb1919.

The original epitopes for mAb2502 and mAb1919 were taken from the manufacturer's data, this has been clarified in the legend to extended data figure 2.

In the legend to figure 3, line 270, I think there may be a mix up. It says that an alternative representation, where the C-terminal region (red) could be the inner core, is on the left of panel C (ii), but in the figure it is shown on the right.

The legend has now been corrected to indicate it's the right panel in figure Cii.

In figure 7, in the bottom panel showing where the integrin headpiece seems to be binding to microfibrils, I would suggest increasing the size of the arrows indicating the binding site.

We have increased the size of the arrow as suggested.

In figure 8, there's mention of domains cbEGF43-45 (line 473), which should be corrected (changed to cbEGF41-43) as fibrillin only has 43 cbEGF domains.

This has been corrected in the legend and figure to cbEGF41-43

Final Decision Letter:

Message 28th Feb 2023

:

Dear Prof. Baldock,

We are now happy to accept your revised paper "Fibrillin microfibril structure identifies long-range effects of inherited pathogenic mutations affecting a key regulatory latent TGF β -binding site" for publication as a Article in Nature Structural & Molecular Biology.

Your paper will be published online soon after we receive proof corrections and will appear in print in the next available issue. You can find out your date of online publication by contacting the production team shortly after sending your proof corrections. Content is published online weekly on Mondays and Thursdays, and the embargo is set at 16:00 London time (GMT)/11:00 am US Eastern time (EST) on the day of publication. Now is the time to inform your Public Relations or Press Office about your paper, as they might be interested in promoting its publication. This will allow them time to prepare an accurate and satisfactory press release. Include your manuscript tracking number (NSMB-A45775B) and our journal name, which they will need when they contact our press office.

About one week before your paper is published online, we shall be distributing a press release to news organizations worldwide, which may very well include details of your work. We are happy for your institution or funding agency to prepare its own press release, but it must mention the embargo date and Nature Structural & Molecular Biology. If you or your Press Office have any enquiries in the meantime, please contact press@nature.com.

Please note that *Nature Structural & Molecular Biology* is a Transformative Journal (TJ). Authors may publish their research with us through the traditional subscription access route or make their paper immediately open access through payment of an article-processing charge (APC). Authors will not be required to make a final decision about access to their article until it has been accepted. https://www.nature.com/protocolexchange/about

[href="https://www.springernature.com/gp/open-research/transformative-journals">](https://www.springernature.com/gp/open-research/transformative-journals) Find out more about Transformative Journals

Authors may need to take specific actions to achieve compliance with funder and institutional open access mandates. If your research is supported by a funder that requires immediate open access (e.g. according to Plan S principles) then you should select the gold OA route, and we will direct you to the compliant route where possible. For authors selecting the subscription publication route, the journal's standard licensing terms will need to be accepted, including self-archiving policies. Those licensing terms will supersede any other terms that the author or any third party may assert apply to any version of the manuscript.

Sincerely,

Dimitris Typas
Associate Editor
Nature Structural & Molecular Biology
ORCID: 0000-0002-8737-1319

Click here if you would like to recommend Nature Structural & Molecular Biology to your librarian:

<http://www.nature.com/subscriptions/recommend.html#forms>